# Phospholamban antisense oligonucleotides improve cardiac function in murine cardiomyopathy

Niels Grote Beverborg [1,11], Daniela Später [2,3,11 ✉], Ralph Knöll [2,3,11], Alejandro Hidalgo[2,3,9,10], Steve T. Yeh[4], Zaher Elbeck [3], Herman H. W. Silljé [1], Tim R. Eijgenraam [1], Humam Siga[3], Magdalena Zurek[2], Malin Palmér[2,5], Susanne Pehrsson[2], Tamsin Albery[2], Nils Bomer [1], Martijn F. Hoes [1], Cornelis J. Boogerd [6], Michael Frisk [7], Eva van Rooij [6], Sagar Damle[4], William E. Louch[7], Qing-Dong Wang [2], Regina Fritsche-Danielson[2], Kenneth R. Chien [3,8], Kenny M. Hansson[2], Adam E. Mullick [4,12 ✉], Rudolf A. de Boer [1,12] & Peter van der Meer [1,12 ✉]

Heart failure (HF) is a major cause of morbidity and mortality worldwide, highlighting an urgent need for novel treatment options, despite recent improvements. Aberrant $Ca^{2+}$ handling is a key feature of HF pathophysiology. Restoring the $Ca^{2+}$ regulating machinery is an attractive therapeutic strategy supported by genetic and pharmacological proof of concept studies. Here, we study antisense oligonucleotides (ASOs) as a therapeutic modality, interfering with the PLN/SERCA2a interaction by targeting *Pln* mRNA for downregulation in the heart of murine HF models. Mice harboring the PLN R14del pathogenic variant recapitulate the human dilated cardiomyopathy (DCM) phenotype; subcutaneous administration of PLN-ASO prevents PLN protein aggregation, cardiac dysfunction, and leads to a 3-fold increase in survival rate. In another genetic DCM mouse model, unrelated to PLN (*Cspr3/Mlp* $^{-/-}$), PLN-ASO also reverses the HF phenotype. Finally, in rats with myocardial infarction, PLN-ASO treatment prevents progression of left ventricular dilatation and improves left ventricular contractility. Thus, our data establish that antisense inhibition of PLN is an effective strategy in preclinical models of genetic cardiomyopathy as well as ischemia driven HF.

[1] Department of Cardiology University Medical Center Groningen, University of Groningen, Groningen, The Netherlands. [2] Bioscience Cardiovascular, Research and Early Development, Cardiovascular, Renal and Metabolism (CVRM), BioPharmaceuticals R&D, AstraZeneca, Gothenburg, Sweden. [3] Integrated Cardio Metabolic Center (ICMC), Karolinska Institutet, Huddinge, Sweden. [4] Ionis Pharmaceuticals, Carlsbad, CA, USA. [5] Laboratory of Experimental Biomedicine, Core Facilities, Sahlgrenska Academy, Gothenburg University, Göteborg, Sweden. [6] Department of Molecular Cardiology, Hubrecht Institute, Royal Netherlands Academy of Arts and Sciences (KNAW), University Medical Center Utrecht, Utrecht, The Netherlands. [7] Institute for Experimental Medical Research, Oslo University Hospital and KG Jebsen Center for Cardiac Research, University of Oslo, Oslo, Norway. [8] Department of Cell and Molecular Biology (CMB), Karolinska Institute, Stockholm, Sweden. [9] Present address: Murdoch Children's Research Institute (MCRI), Flemington, Melbourne, VIC, Australia. [10] Present address: Department of Paediatrics, University of Melbourne, Melbourne, VIC, Australia. [11] These authors contributed equally: Niels Grote Beverborg, Daniela Später, Ralph Knöll. [12] These authors jointly supervised this work: Adam E. Mullick, Rudolf A. de Boer, Peter van der Meer. ✉ email: daniela.spaeter@astrazeneca.com; AMullick@ionisph.com; p.van.der.meer@umcg.nl

Despite advances in treatment, HF patients have poor long-term outcomes with 5-year mortality rates of 50%, exceeding common forms of cancer such as prostate and breast cancer[1–3]. The need for a therapy capable of addressing the root cause of cardiac dysfunction in HF remains high[4]. One of the key features of HF is aberrant $Ca^{2+}$ cycling, which impairs cardiac muscle contraction, and is associated with cardiac arrhythmias and adverse remodeling[5–7]. A critical contributor is reduced activity of the sarcoplasmic reticulum $Ca^{2+}$-ATPase (SERCA2a), accompanied by a relative increase of unphosphorylated phospholamban (PLN), a SERCA2a inhibitor[8–10]. PLN is a principal regulator of SERCA2a activity, maintaining appropriate sarcoplasmic reticulum $Ca^{2+}$ cycling, which has been shown critical for cardiac relaxation and contraction[11]. Evidence based HF treatments primarily target neurohormonal systems, while the assumed crucial calcium handling proteins are left untouched. There have been attempts to target SERCA2a and its partners. To date, most efforts have focused on increasing SERCA2a expression using adeno-associated virus based gene therapy in multiple preclinical models of HF, including aortic-banded rats[12], volume overloaded pigs[13], and cardiomyocytes from failing human hearts[14] and also clinical trials[15]. In preclinical studies, SERCA2a overexpression prevented cardiac dysfunction or improved cardiac function; however, the CUPID 2 trial, assessing AAV1 mediated SERCA2a gene delivery in human patients with HF failed to show any clinical benefit[15]. In a selective subset of patients with deteriorating disease, no adequate levels of vector DNA were detected, providing a possible explanation for the lack of efficacy[15]. An alternative strategy is to target PLN instead. Excessive PLN mediated inhibition of SERCA2a activity is suggested to be an important contributor to the pathogenesis of HF. This is illustrated by studies demonstrating improved cardiac function and mitigation of HF following genetic manipulation of PLN[8,16]. Genetic depletion of PLN rescued cardiac function in a genetic mouse model lacking muscle LIM protein (Cspr3/Mlp $^{−/−}$)[8]. Expression of non-inhibitory PLN (point-mutants or pseudo-phosphorylated PLN) has also been shown to improve contractility in vitro and in cardiomyopathic hamsters[8,16]. Importantly, human PLN variants, such as the deletion of arginine at location 14 (R14del), are associated with severe forms of cardiomyopathy[17,18].

The PLN/SERCA2a interaction has been difficult to target by conventional small-molecule drugs[19]. Thus, the PLN/SERCA2a axis has remained an unfulfilled promise as a therapeutic target. Antisense oligonucleotides (ASOs) provide the possibility to dose-dependently inhibit protein expression without the need for genetic manipulation. ASOs have recently achieved clinical success in ameliorating diseases such as spinal muscular atrophy, transthyretin (TTR) amyloidosis, and hyperlipidemia[20]. Improvements in ASO potency and delivery through advances in medicinal chemistry offer the potential to target tissues beyond the liver such as skeletal muscle and heart[21–23]. A previous report showed cardiac PLN downregulation and short-term improvements in cardiac contractility following hydrodynamic intravenous delivery of a locked nucleic acid (LNA)-modified gapmer ASO targeting PLN in mice with pressure overload induced HF[23]. With such recent advances, we set out to more rigorously design, screen and evaluate mouse- and rat-specific gapmer ASOs containing constrained ethyl (cET)-modifications. In this work, we show that ASOs targeting PLN decrease cardiac Pln mRNA expression resulting in significant PLN protein reductions. This prevents or reverses cardiac dysfunction in three different rodent models of HF and cardiomyopathy, and significantly increases lifespan in a model of hereditary PLN R14del cardiomyopathy.

## Results

### Screening identifies a highly potent PLN-ASO which improves diastolic $Ca^{2+}$ uptake and relaxation in vitro.

In silico, in vitro, and in vivo analyses of prospective ASOs targeting mouse Pln were used to identify 2 lead candidates for in vivo studies (Fig. 1a). To determine the rank order of prospective leads, mouse primary cardiomyocytes were treated with 0.56, 1.67, 5, or 15 μM of ASO for 24 h. ASOs that produced dose-dependent reductions in Pln mRNA with at least a 50% reduction at 5 μM or had potent IC50 responses were considered for in vivo evaluation (Fig. 1b). After an additional selection based on tolerability prediction using a proprietary in silico prediction algorithm, 20 PLN-ASOs were selected for evaluation in healthy C57BL/6 mice. ASOs were evaluated as their 5'-palmitic acid conjugates (denoted by "_C"), which have previously been shown to improve heart distribution and activity in healthy mice[21,22]. ASOs were evaluated following three subcutaneous (s.c.) administrations of 50 mg/kg given on days 0, 5 and 8, with blood and tissues collected on day 13. ASO#26_C reduced heart Pln expression by >50% without adverse effects on plasma alanine aminotransferase (ALT) or aspartate transaminase (AST) and was selected as a lead for further studies (Fig. 1c). A subsequent assessment in healthy mice demonstrated potent dose–responsive reductions in cardiac Pln mRNA of up to ~80% with 100 mg/kg without any increase in plasma ALT (Fig. 1d). Another candidate, ASO#27_C, showed the most potent heart Pln reductions but also evoked non-significant elevations in plasma ALT and AST (Fig. 1c). As 5'-palmitic acid conjugation of ASOs has been shown to also increase ASO liver accumulation[21] we attempted to mitigate the liver toxicity of ASO#27_C by comparing the non-palmitated version (ASO#27) to ASO#26_C. Mice were dosed with 3 administrations given on day −17, −10, and −3. Tissues were harvested on day 0, 14, 28 and 42. ASO#26_C and ASO#27 were dosed at 50 and 100 mg/kg, respectively. Similar levels of cardiac Pln inhibition were observed following either treatment with acceptable levels of tolerability (Fig. 1e and Supplementary Fig. 1). Functional effects of Pln mRNA downregulation by ASO#27 were assessed in healthy neonatal mouse cardiomyocytes. After 30 h of exposure to 15 μM PLN-ASO#27, an increased maximal $Ca^{2+}$ flux and decreased half time of both diastolic $Ca^{2+}$ reuptake and physical relaxation as compared to PBS were observed (Fig. 1f). Despite an increased maximal $Ca^{2+}$ flux, no increase in fractional shortening (FS) was seen. For subsequent experiments, both ASO#26_C and ASO#27 were chosen as lead ASOs and were further evaluated in murine HF models as described below.

### PLN-ASO treatment prevents cardiac dysfunction and mortality in a PLN R14del mouse model.

To assess in vivo functionality of the identified PLN-ASO leads, we turned to a model of genetic cardiomyopathy caused by a pathogenic variant of PLN. Several PLN mutations are known to cause cardiomyopathy in humans, with the deletion of arginine at position 14 of the PLN protein (R14del) being one of the most prevalent variants which leads to severe HF and malignant ventricular arrhythmias, and has recently been increasingly studied[18,24]. Although the direct effects on $Ca^{2+}$ cycling are part of ongoing discussion, the PLN R14del pathogenic variant is reported to result in PLN protein aggregates, myocardial fibrosis and cardiac dysfunction[25,26]. We reasoned that ASO induced reductions of PLN protein expression has the potential to prevent disease development and progression, thereby representing a possible precision medicine approach to treat PLN R14del induced cardiomyopathy. A mouse model carrying the PLN R14del pathogenic variant in both alleles (PLN R14$^{Δ/Δ}$) recapitulated all common features of the human

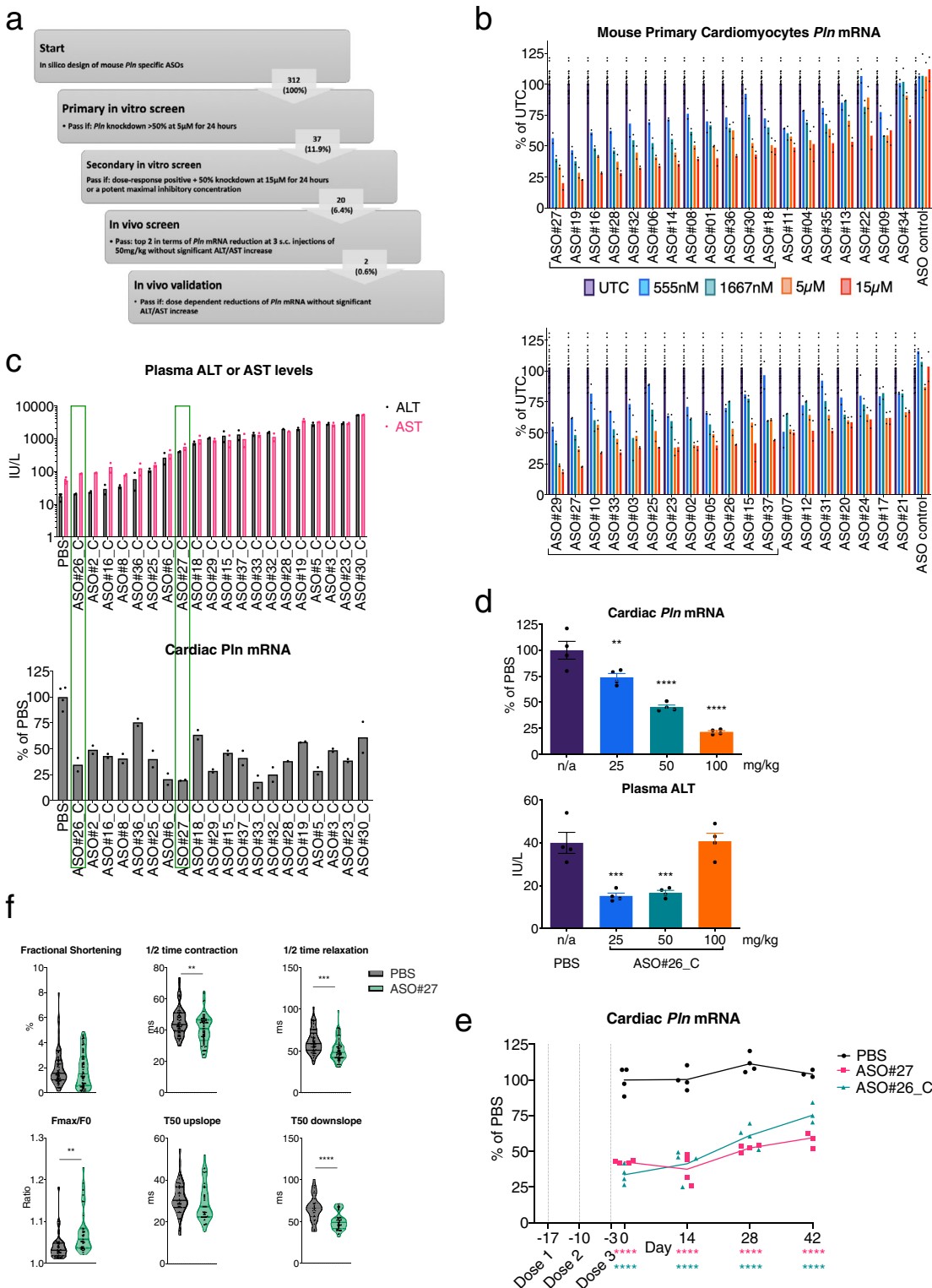

phenotype observed in heterozygous carriers, yet in an accelerated fashion, with rapid development of DCM, myocardial fibrosis, and PLN protein aggregates resulting in premature death at the age of 8–9 weeks[27]. This phenotype is observed despite endogenous downregulation of PLN RNA and protein expression in untreated PLN R14$^{\Delta/\Delta}$ mice compared to wild-type mice[27]. We hypothesize this phenotype to be caused by the residual expression of mutant PLN, we aim to further reduce this expression utilizing the PLN-ASO.

As the PLN R14$^{\Delta/\Delta}$ mice previously showed increased cardiac dimensions and a reduced left ventricular ejection fraction (LVEF) at 6 weeks of age, we started our 4-week dosing regimen of PLN-ASO#27 at 3 weeks of age (Fig. 2a). This resulted in 42% reduction of cardiac *Pln* mRNA (±6%, *P* value = 0.0153, Fig. 2b) and 50% reduction of total (both soluble and insoluble/ aggregated) PLN protein levels (±18%, *P* value = 0.0142) at treatment week 4 (T4) and 7 weeks of age (Fig. 2c and Supplementary Fig. 2). *Atp2a2* mRNA and SERCA protein levels

**Fig. 1 A large in vitro and in vivo screen resulting in a PLN ASO that dose-dependently decreases cardiac *Pln* mRNA and improves in vitro calcium fluxes. a** Schematic illustration of experimental steps leading to identification of 2 mouse PLN ASO leads for in vivo studies. **b** Results of secondary in vitro screen illustrating *Pln* mRNA reductions in primary mouse cardiomyocytes in a dose–response comparison. ASO candidates were selected (20) based on demonstrated dose–response and >50% *Pln* mRNA reduction after 24 h of treatment, $N = 24$ for untreated control (UTC) and $N = 2$ for each PLN ASO. **c** Selection of 2 ASO leads from in vivo screen in healthy C57BL/6 mice based on heart *Pln* mRNA reductions and plasma ALT and AST after 3 s.c. injections of 50 mg/kg, $N = 4$ for PBS and $N = 2$ for each PLN ASO. **d** In vivo dose–response reductions of cardiac *Pln* mRNA by PLN-ASO#26_C (s.c. administration of 3 × 25, 50 or 100 mg/kg, analysis after 2 weeks) and plasma analysis for ALT levels, $N = 4$ per group. **e** Comparable sustained cardiac *Pln* mRNA downregulation achieved with 3 × 50 mg/kg (day −17, −10, and −3) palmitate PLN-ASO#26_C and 3 × 100 mg/kg (day 0, 7, and 14) nonconjugated PLN-ASO#27. Protein analysis was performed on day 17, 31, 45, and 56, $N = 4$ per group. **f** In vitro response to 30 h exposure to 15 μM ASO#27 or PBS. Neonatal mouse cardiomyocytes were cultured on micropatterned flexible substrates in islands. Fractional shortening was determined using the BASiC technique ($N = 51$ for PBS and $N = 57$ for PLN ASO), and 20 min Fluo-4 incubation was performed to assess calcium cycling ($N = 34$ for PBS [$N = 33$ for T50 downslope] and $N = 31$ for PLN ASO). Violin plots with individual dots and median and interquartile range or histograms with standard error of the mean (SEM) are depicted. Students' *T* test and one-way analyses of variance were used for statistical analysis. Asterix denotes significance level based on two-sided *P* values compared to PBS with: *$P$ value < 0.05; **$P$ value < 0.01; ***$P$ value < 0.001; ****$P$ value < 0.0001. ASO antisense oligonucleotide, ALT Alanine transaminase, AST Aspartate transaminase, UTC untreated control, Fmax maximal fluorescence, F0, minimal fluorescence. Source data are provided as a Source Data file.

remained unchanged (Supplementary Figs. 3–6). Combined, this resulted in a change of the SERCA2/PLN ratio in favor of SERCA2 (Supplementary Figs. 5, 6). Immunofluorescence showed a lower abundance of PLN protein aggregates in PLN-ASO treated mice (Fig. 2d). Additionally, RIPA-insoluble protein fractions of PLN R14$^{\Delta/\Delta}$ mice hearts contained more PLN protein as compared to wild-type mice, indicating aggregated PLN protein complexes (Supplementary Figs. 3–5). In PLN-ASO treated mice hearts, RIPA-insoluble PLN protein levels were 89% lower than values in vehicle control (±12%, Supplementary Figs. 4, 5, $P$ value < 0.0001). Strikingly, magnetic resonance imaging (MRI) of the heart at 7 weeks of age revealed almost completely normal cardiac dimensions and function in PLN-ASO treated mice, whereas severe DCM was observed in vehicle-treated mice (Fig. 2e, f, LVEF 53 ± 2 vs. 27 ± 2%, $P$ value < 0.0001, left ventricular end-diastolic volume [LVEDV] 41 ± 2 vs. 54 ± 2.0 mL, $P$ value < 0.0001 and left ventricular end-systolic volume [LVESV] 20 ± 1 vs. 39 ± 2 mL, $P$ value < 0.0001). No changes in heart rate were observed on ECG (Supplementary Fig. 7). Also, no signs of treatment toxicity were observed based on plasma AST and ALT, haematoxylin and eosin (H&E) staining of kidney, liver or spleen sections (Supplementary Fig. 7).

Genome wide transcriptome profiling by RNA-Sequencing revealed that PLN R14$^{\Delta/\Delta}$ mutant hearts displayed marked differences in global gene expression compared to wild-type control hearts (Fig. 2g). Moreover, PLN-ASO treatment attenuated gene expression changes in mutant hearts, thereby indicating that PLN-ASO treated hearts more closely resemble wild-type hearts than the PBS treated mutant counterparts. When assessing the functions of genes mostly contributing to the difference between PLN R14$^{\Delta/\Delta}$ and controls (principle component 1; PC1, and genes differentially expressed in PLN R14$^{\Delta/\Delta}$ vs. wild-type and subsequently reversed by PLN-ASO vs. PBS), we observed increased expression of genes involved in fatty acid metabolism, and decreased expression of genes involved in protein processing in the endoplasmic reticulum and extracellular matrix, providing further confirmation that key determinants of cardiomyopathy such as energy metabolism and fibrosis are attenuated by the PLN-ASO treatment (Fig. 2g and Supplementary Fig. 8a–c).

Another typical hallmark of PLN R14del cardiomyopathy in humans is a low voltage ECG, which was also observed in PLN R14$^{\Delta/\Delta}$ mice (Fig. 2h, i, expressed by low R-amplitude), and was partially restored by PLN-ASO treatment (0.84 ± 0.05 vs. 0.55 ± 0.05 mV, $P$ value < 0.0001). We excluded known confounders of a low voltage ECG including obesity (body weight to tibia length ratio 1.14 ± 0.05 vs. 1.46 ± 0.02, $P$ value = 0.0012, of PLN R14$^{\Delta/\Delta}$ vs. wild-type, Supplementary Fig. 9), and cardiac effusion

which was not observed at sacrifice in PLN R14$^{\Delta/\Delta}$ mice. Masson's trichrome staining revealed a significant prevention of myocardial fibrosis in PLN-ASO treated mice (Fig. 2j, k, fold change compared to WT 2.4 ± 0.4 vs. 17.17 ± 4.1%, $P$ value = 0.0032). The significantly higher ECG voltages seen after 4 weeks of PLN-ASO treatment, are therefore arguably explained by the prevention of fibrosis and PLN protein aggregation.

Since cardiac function was almost entirely preserved, we investigated the effects on lifespan. After an initial loading phase with a high-dose (100 mg/kg from 3 to 6 weeks of age), mice were treated with a lower maintenance dose (50 mg/kg every 4 weeks, Fig. 2a). This increased lifespan by over 3-fold from the average of 7.5 weeks to at least 24 weeks (Fig. 2l, log-rank $P$ value < 0.0001), with only 1 mouse dying from HF before study termination at the age of 26 weeks. To gain insight into the development and progression of the disease phenotype and to identify disease contributors, a set of mice were sacrificed every 4 weeks during the late study phase (follow-up). Age-matched vehicle-treated control PLN R14$^{\Delta/\Delta}$ mice were not available for these analyses as they all died before the age of 9 weeks, which hampers accurate interpretation of follow-up PLN mRNA and protein analyses in PLN-ASO treated mice. Based on known PLN ASO pharmacokinetics, this dosing regime was chosen not to keep PLN mRNA and protein levels knocked down at steady state, but to allow a slow incline of PLN levels over time and therefore being insufficient to prevent disease progression. PLN-ASO treated PLN R14$^{\Delta/\Delta}$ mice slowly progressed to HF, with first signs of cardiac disease at T19, which corresponds to the age of 22 weeks (Supplementary Fig. 10).

We correlated global transcriptome data with cardiac function (LVEF) acquired just before sacrifice at different time points during the study to identify possible pathways associated with a positive treatment effect. Gene ontology analysis revealed oxidative phosphorylation and other metabolic pathways and protein processing in the endoplasmic reticulum as the top affected pathways (Supplementary Fig. 8d, e). Additionally, we observed a durable downregulation of genes involved in the unfolded protein response in PLN-ASO treated PLN R14$^{\Delta/\Delta}$ mice (Supplementary Fig. 8f). These findings provide further support to the idea that a reduction of toxic PLN protein aggregates by PLN-ASO might contribute to the significant benefits observed after treatment. Sarcoplasmic reticulum protein processing is dependent on local Ca$^{2+}$ levels[28], but no further indication of altered Ca$^{2+}$ related pathways was observed. In summary, PLN-ASO treatment slowed down the development of cardiac fibrosis, protein aggregation and DCM in PLN R14$^{\Delta/\Delta}$ mice and extended survival over threefold.

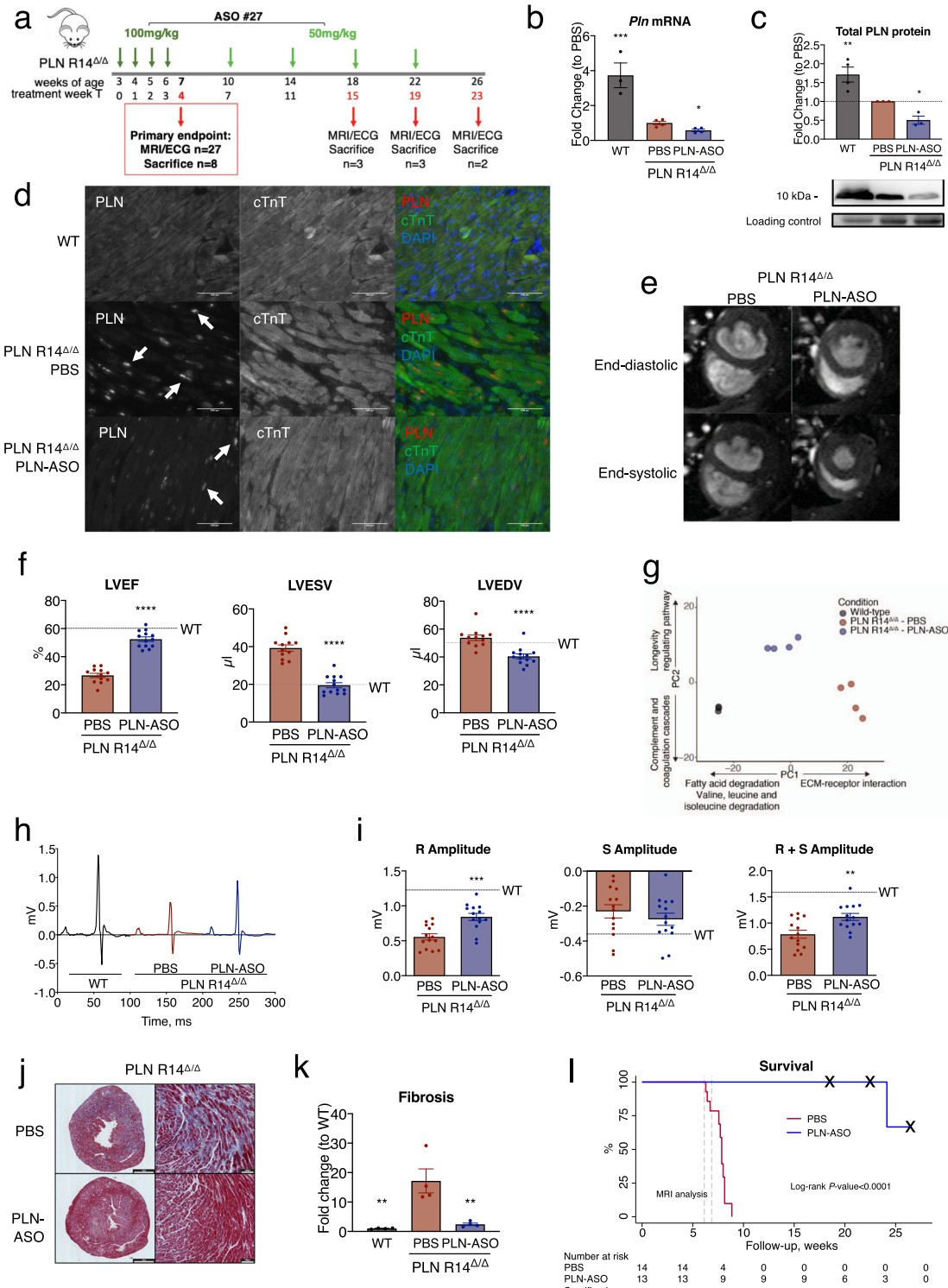

**PLN-ASO treatment reverses signs and symptoms of heart failure in Cspr3/Mlp$^{-/-}$ mice.** In PLN R14$^{Δ/Δ}$ mice, PLN-ASO directly addressed the root cause of the disease, resulting in a dramatic treatment benefit. Therefore, we were also interested in assessing PLN-ASO effects in a cardiac disease model where calcium handling is impaired, but where PLN is not believed to be the direct instigator of disease. To this end, we evaluated mice lacking muscle LIM protein (Cspr3/Mlp$^{-/-}$), an established model for DCM[29]. CSPR3/MLP is a structural protein located at the Z-line in cardiac and slow-twitch skeletal muscle cells, but

also has a nuclear translocation signal. It has been implicated to play a role in muscle differentiation and as a mechanical stress sensor converting upstream signals into both intra- and intercellular functional responses[30]. Mutations in this gene can lead to both DCM and familial hypertrophic cardiomyopathy in humans[31]. Cspr3/Mlp$^{-/-}$ mice develop a DCM phenotype, typically at the age of 8–12 weeks; however, disease onset variability was observed[29]. Previous work has established this model to be amenable to improvements in cardiac function by additional genetic ablation of Pln[8]. Recently, this has been confirmed in a

**Fig. 2 Repeated subcutaneous injection of PLN ASO in mice with PLN R14$^{\Delta/\Delta}$ prevents cardiac dysfunction and improves survival. a** Experimental design of the PLN R14$^{\Delta/\Delta}$ intervention study, consisting of ASO#27 treatment with a 4-week loading phase with weekly dosing of 100 mg/kg and a 20-week maintenance phase with dosing of 50 mg/kg once every 4 weeks. Data points for functional and biochemical assessments were obtained at treatment week 4 (T4) which was the primary endpoint, 15 (T15), 19 (T19), and 23 (T23) at the end of the study. **b** SYBR green qPCR results of cardiac *Pln* mRNA expression ($n = 3$ for wild-type [WT] and $n = 4$ for PBS and PLN-ASO). **c** Semi-quantified western blot analysis of PLN protein expression using LV protein lysates. Intensities were normalized to total protein loading control and the PBS treated mice ($n = 3$ per group). **d** Immunofluorescence of WT, PBS treated and PLN-ASO treated PLN R14$^{\Delta/\Delta}$ mice after 4 weeks treatment. Sections were co-stained for cardiac Troponin T (green), PLN (red), and DAPI (blue). Note the lower presence of visible PLN aggregations (white arrows) and the higher structural integrity of the PLN-ASO treated hearts vs. PBS control. Stainings were performed in two animals per group. **e** Representative MRI images after 4 weeks of treatment. **f** Quantification of LVEF (left ventricular ejection fraction), LVESV (left ventricular end-systolic volume), and LVEDV (left ventricular end-diastolic volume) ($n = 12$ for PBS T4, $n = 13$ for PLN-ASO T4). Wild-type data were previously published, and the average is presented here for reference. **g** Principle component analysis plot of the first 2 principle components derived from the RNA-sequencing of mice left ventricles. **h** Representative ECG tracing of mice after 4 weeks of treatment. **i** Quantification of R, S, and R + S amplitude ($n = 14$ for T4) Wild-type data were previously published and the average is presented here for reference. **j** Representative Masson's trichrome staining after 4 weeks of treatment of PBS and PLN-ASO treated PLN R14$^{\Delta/\Delta}$ cardiac sections. Stainings were performed in four animals per group. **k** Quantification of myocardial fibrosis based on Masson's trichrome staining whereby the blue, fibrotic area is presented as the fold change compared to wild-type in percentage of the total tissue slice surface ($n = 4$ for all groups). **l** Kaplan–Meier survival plot of PLN R14$^{\Delta/\Delta}$ animals treated with PLN-ASO ($n = 13$) and PBS ($n = 14$). Three PBS treated animals died during MRI imaging, the other PBS treated animals died at around 8 weeks of age. The "x" marks the 3-time points at which animals were sacrificed for tissue analyses. One-way analysis of variance was used for statistical analyses, with PBS treated animals as the reference group in multiple comparison analyses. Asterix denotes significance level based on two-sided *P* values compared to PBS with: *P value < 0.05; **P value < 0.01; ***P value < 0.001; ****P value < 0.0001. Single values are depicted, and error bars represent the standard error of the mean (SEM). ASO antisense oligonucleotide, MRI magnetic resonance imaging, and ECG electrocardiogram. Source data are provided as a Source Data file.

study showing that DWORF overexpression, a micropeptide enhancing SERCA2a activity by displacing PLN, prevents cardiac dysfunction in *Cspr3/Mlp$^{-/-}$* mice[32].

To test whether therapeutic reductions of PLN protein levels would improve cardiac function in already diseased mice, PLN-ASO (ASO#26_C or ASO#27) or vehicle control (PBS or control ASO) were subcutaneously administered to *Cspr3/Mlp$^{-/-}$* mice after onset of disease (Fig. 3a, Supplementary Data 1). Mice were followed for 28 days before terminal analyses. PLN-ASO treatment resulted in a 70–90% reduction in *Pln* mRNA (Fig. 3f) and total PLN protein levels (both pentamer and monomer) in the heart (Fig. 3b and Supplementary Fig. 11). Expression of SERCA2 was not significantly affected by the disease condition or PLN-ASO treatment at the mRNA or protein level, while such treatment increased the ratio of SERCA2 to its inhibitor PLN (Fig. 3b, f and Supplementary Fig. 11). PLN protein down-regulation appeared uniformly across cardiomyocytes (Supplementary Fig. 12). A PLN ASO pharmacokinetics and dynamics study showed an almost immediate onset of PLN downregulation on the mRNA level, followed by protein levels around 7 days later, and a sustained PLN knockdown effect post treatment (Supplementary Fig. 13).

Echocardiography was performed at baseline and after 4 weeks of treatment to compare responses to treatment, and mice with signs of HF at study initiation (LVEF ≤ 45%; median, 37.6%; interquartile range, 32.2–42.0%) were analyzed. Improvements of LVEF (60 ± 8 vs. 46 ± 12%), LVESV (31 ± 11 vs. 56 ± 21 mL) and LVEDV (79 ± 22 vs. 100 ± 25 mL) were observed in PLN-ASO treated versus control mice respectively (all *P* value < 0.001, Fig. 3c, d, Supplementary Data 1). Hemodynamic baseline measurements of WT (*Cspr3/Mlp$^{+/+}$*) and *Cspr3/Mlp$^{-/-}$* mice, either PBS or PLN-ASO treated (Supplementary Data 1), showed that left ventricular contractility (assessed by LV dP/dtmax) was significantly enhanced by PLN-ASO treatment compared to *Cspr3/Mlp$^{-/-}$* control mice. In addition, an impaired diastolic function was detected in *Cspr3/Mlp$^{-/-}$* mice (assessed by lower LV dP/dtmin and increased Tau) which was rescued by PLN-ASO treatment to functional levels comparable to wild-type mice (Fig. 3e and Supplementary Fig. 14). In response to increasing doses of the beta-adrenergic agonist dobutamine, *Cspr3/Mlp$^{-/-}$* mice showed a blunted response in contractility and relaxation

compared to littermate wild-type controls, which suggested a reduced cardiac reserve in *Cspr3/Mlp$^{-/-}$* mice. The contractility response was mostly normalized by PLN-ASO treatment, and relaxation was enhanced but not normalized upon increasing dobutamine doses (Fig. 3e). Heart rate was not affected by PLN-ASO treatment (Supplementary Fig. 14). Furthermore, there was no effect on peak systolic and end-diastolic LV pressure (Supplementary Fig. 14) and no effect on heart and body weight (Supplementary Fig. 15). Whole cell Ca$^{2+}$ flux recordings of adult cardiomyocytes isolated from *Cspr3/Mlp$^{-/-}$* mice after 4 weeks of treatment with PLN-ASO versus control in vivo (Fig. 3a), showed significantly enhanced amplitude and upstroke- and decay velocity, without changes in Ca$^{2+}$ flux durations due to pacing at 1 Hz (Supplementary Fig. 16). Finally, RT-PCR analysis of cardiac transcripts demonstrated reversal of HF associated gene expression, such as *Myh7*, *Acta1*, *Nppa*, *Nppb* in *Cspr3/Mlp$^{-/-}$* mice upon PLN-ASO treatment (Fig. 3f). A reduction of ANP levels was also observed in plasma samples (Fig. 3f). A validation study was performed comparing PLN-ASO (ASO#26_C) to a control ASO, using a similar design as the other studies (Supplementary Fig. 17). PLN-ASO significantly improved LVEF and LVESV over the control ASO treated mice. In summary, PLN ASO treatment substantially reversed the DCM phenotype of *Cspr3/Mlp$^{-/-}$* mice towards that of healthy mice.

**PLN-ASO treatment improves contractility and reduces cardiac dimensions after myocardial infarction.** Acquired HF is the most prevalent form of cardiomyopathy in humans. Therefore, we aimed to test the PLN-ASO treatment effects in an acquired HF model. For this purpose, we employed the rat post myocardial infarction (MI) model, which is widely used to test potential therapeutic interventions to treat human disease. In this model, the ischemic event triggers massive cell loss, while cardiac function progressively declines, accompanied by pathological remodeling and impaired intracellular Ca$^{2+}$ handling. This has been extensively evaluated in preclinical MI models and isolated cardiomyocytes from patients with post-MI HF[33,34]. A previous study showed that reduction of inhibitory PLN, and thereby enhanced SERCA2a activity, in post-MI rats could prevent progressive cardiac dysfunction and pathological remodeling[35]. An additional ASO screen for a rat-specific PLN-ASO was

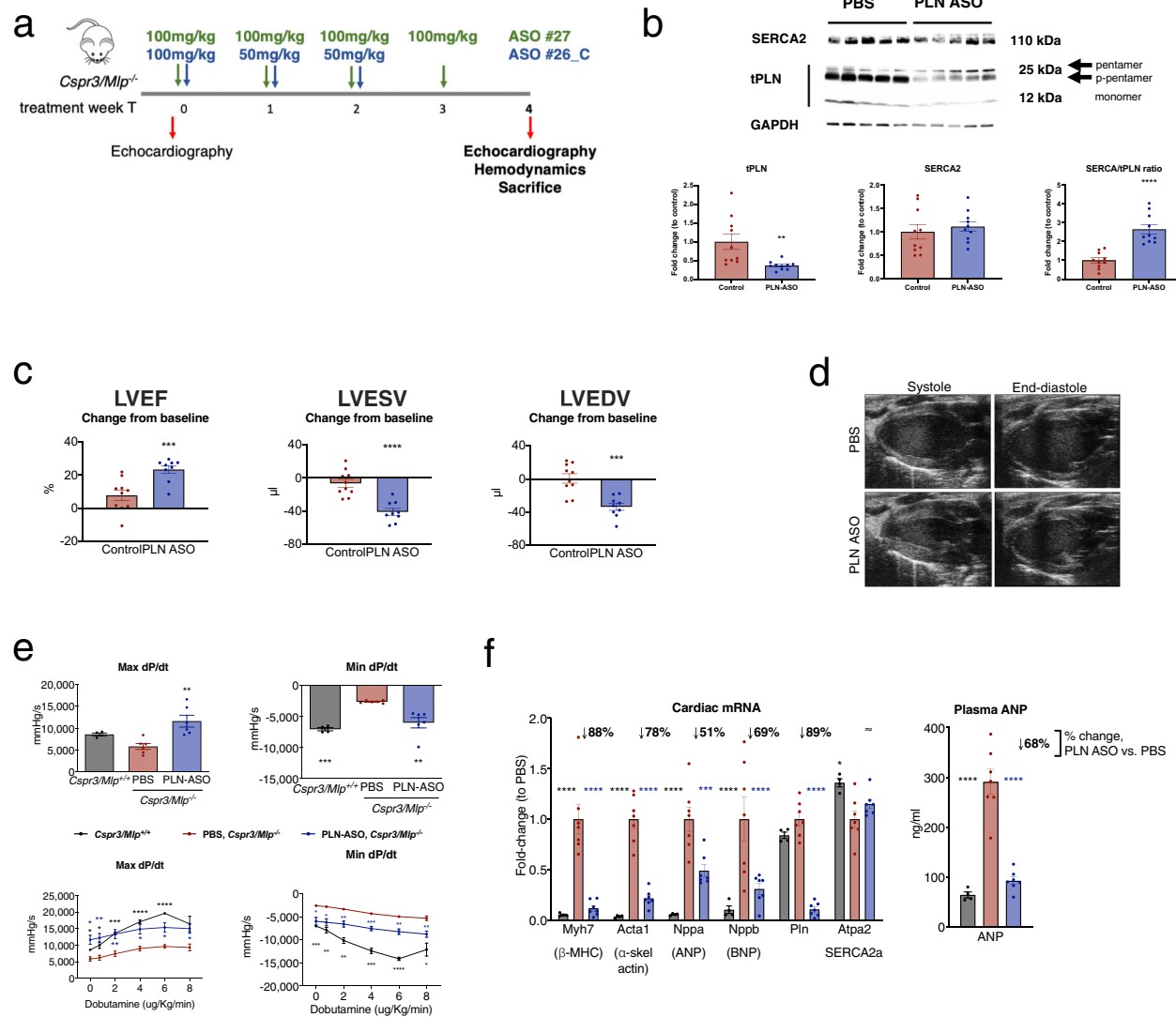

**Fig. 3 Repeated subcutaneous injection of PLN ASO in Cspr3/Mlp⁻/⁻ mice reverses signs and symptoms of heart failure. a** Experimental design of the PLN-ASO *Cspr3/Mlp⁻/⁻* intervention study. Repeated studies were run with either PLN-ASO #27 (100 mg/kg on Days 0, 7, 14, and 21) or PLN-ASO #26_C (100 mg/kg on day 0, 50 mg/kg on days 7 and 14) (Supplementary Data 1). Echocardiography was performed at baseline (i.e., before treatment initiation), and at end of study (i.e., after 28 days of treatment), followed by termination of animals. Hemodynamic assessment was done between day 24 and 28. **b** Representative Western blot of LV protein lysates stained for PLN protein to detect the monomer and pentamer. Protein loading was normalized to GAPDH and total PLN protein were semi-quantified by calculating intensities relative to PBS treated control (PBS $n = 10$, PLN-ASO #26_C $n = 10$). **c** Individual echocardiography assessment and quantification of left ventricular ejection fraction (LVEF), left ventricular end-systolic volume (LVESV), and left ventricular end-diastolic volume (LVEDV) 28 days after treatment relative to baseline measurements. *Cspr3/Mlp⁻/⁻* mice with LVEF ≤ 45% at study initiation were included in analysis, combination of study 1* + 2^ (PLN-ASO #26_C treated $n = 6* + 3$^, PBS treated $n = 7*$, Control-ASO $n = 1$^, and no injection $n = 2$^ (Supplementary Data 1)). **d** Representative B-MODE echocardiography image of a *Cspr3/Mlp⁻/⁻* heart 28 days post treatment with PBS or PLN-ASO#26_C. **e** Hemodynamic assessment of the maximum/minimum first derivative of LV pressure (Max dP/dt and Min dP/dt) was performed at 24–28 days following PLN-ASO_#27 or PBS treatment at baseline and upon increasing dobutamine doses of 0, 2, 4, 6, 8 µg/kg/min in a study 3 ($n = 4$ for *Cspr3/Mlp⁺/⁺*, $n = 6$ *Cspr3/Mlp⁻/⁻* PBS, and $n = 6$ *Cspr3/Mlp⁻/⁻* PLN-ASO_#27. For this study cohort *Cspr3/Mlp⁻/⁻* mice were selected based on high plasma ANP levels (>119 ng/ml) and randomized into treatment groups based on LVEF baseline levels (Supplementary Data 1)). **f** RT-PCR fold change analysis of transcriptional heart failure markers and plasma ANP ELISA in *Cspr3/Mlp⁻/⁻* PBS ($n = 7$) and *Cspr3/Mlp⁻/⁻* PLN-ASO #27 ($n = 7$) mice relative to non-HF *Cspr3/Mlp⁺/⁺* mice ($n = 4$) after 4 weeks of treatment. Students' *T* test, one and two-way analysis of variance were used for analyses, with PBS/vehicle-treated animals as the reference group in multiple comparison analyses. Asterix denotes significance level based on two-sided *P* values compared to PBS/vehicle control with: **P* value < 0.05; ***P* value < 0.01; ****P* value < 0.001; *****P* value < 0.0001. Single values are depicted, and error bars represent standard error of the mean (SEM). ASO antisense oligonucleotide, ANP atrial natriuretic peptide. Source data are provided as a Source Data file.

performed, similar to that in mouse (Supplementary Fig. 18). Rats underwent permanent left anterior descending artery ligation to induce MI, and then a 6-week interim period to allow disease progression. At this time point, rats underwent an MRI to obtain baseline measurements (average LVEF = 46%) and were

randomized into the following treatment groups: PBS, scrambled Control-ASO, PLN-ASO low (25 mg/kg) and high (50 mg/kg) dose, and a repeated dosing regimen was initiated (Fig. 4a). After 5 weeks of treatment, PLN-ASO resulted in a 29% (±12%) or 66% (±12%) reduction of *Pln* mRNA levels ($P = 0.07$ and $P = 0.0001$,

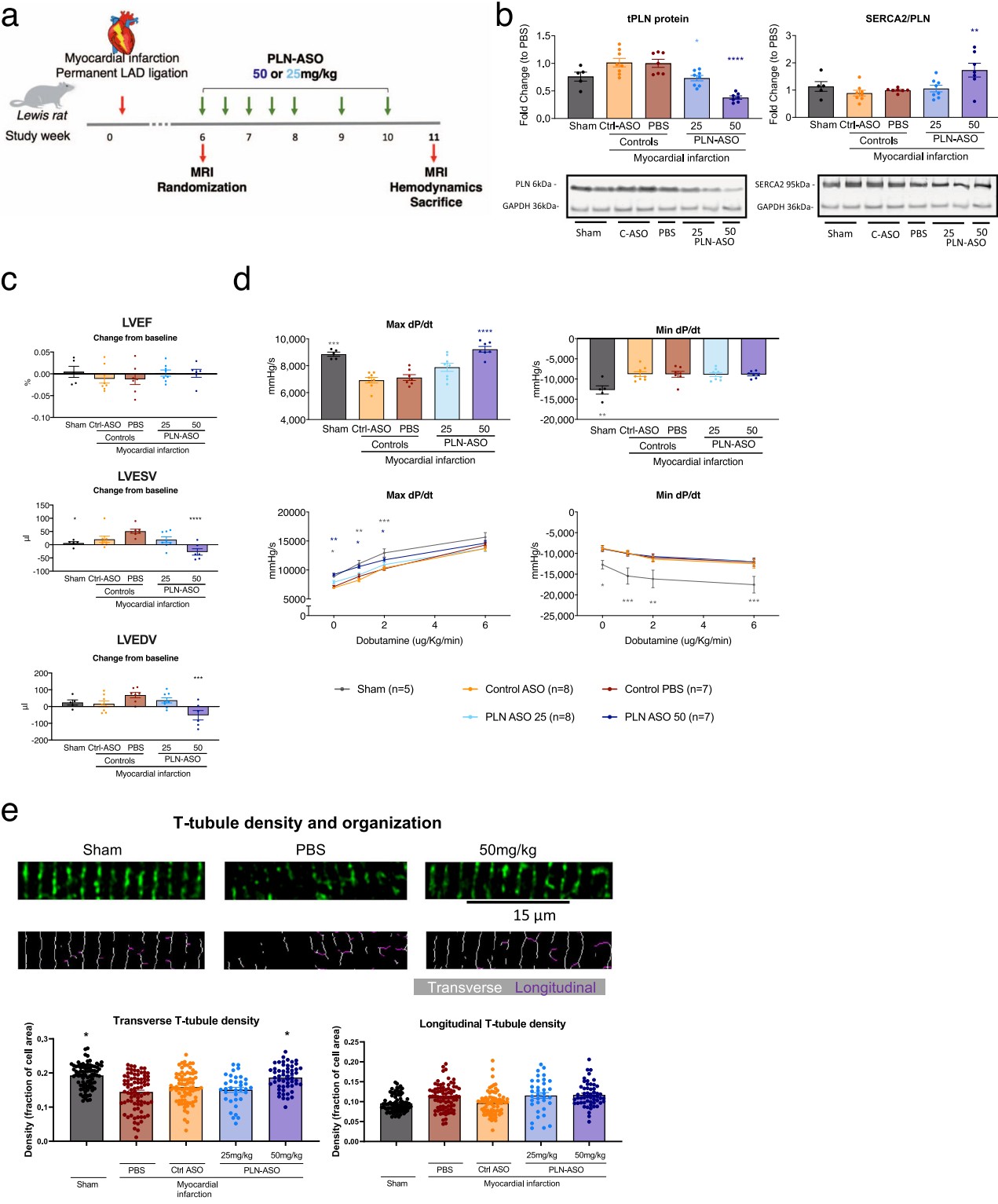

Supplementary Fig. 19), and 30% (±4%) or 57% (±4%) reduction in PLN protein levels (both $P < 0.0001$, Fig. 4b and Supplementary Fig. 20) in the low and high-dose group respectively. SERCA2 protein expression levels were not significantly affected by MI or PLN-ASO treatment, and thus the ratio of SERCA2 to PLN expression was significantly increased in rats treated with 50 mg/kg PLN-ASO (Fig. 4b and Supplementary Figs. 19, 20). MRI functional analyses showed no differences between the two control groups (PBS and scrambled ASO treated rats). The

average LVEF at the start of treatment was reduced in MI compared to sham (46% (±1%) versus 62% (±1%), $P$ value < 0.001), suggesting a moderate HF state, but no further deterioration in LVEF was observed in the following 5-week period in any of the treatment groups (Fig. 4c). However, both LVESV and LVEDV tended to increase with time in the control post-MI groups, and this effect was significantly reversed by high-dose PLN-ASO treatment (change from baseline: LVESV, +51 μl (±8 μl) versus −27 μl (±12 μl), $P$ value < 0.0001, LVEDV, +68 μl (±16 μl) versus

**Fig. 4 Repeated subcutaneous injection of PLN ASO in rats with a myocardial infarction improves improves contractility and reduces cardiac dimensions. a** Experimental design of the rat post myocardial infarction intervention study. Six weeks after myocardial infarction, induced by permanent left anterior descending (LAD) artery ligation, an MRI was performed, and rats were randomized into groups. Treatment of ASO#136 was initiated at a 2× weekly dosing for the first 2 weeks followed by 1× weekly dosing (sham $n = 5$, PBS $n = 7$, Control-ASO $n = 8$ (50 mg/kg), PLN-ASO low dose $n = 8$ (25 mg/kg), and PLN-ASO high-dose $n = 7$ (50 mg/kg)). MRI, invasive hemodynamics and sacrifice was performed 5 weeks after treatment initiation, 11 weeks after myocardial infarction. **b** Western blot results of LV protein lysates stained for PLN and SERCA2 protein and semi-quantified relative to PBS treated control samples, intensities were normalized to GAPDH ($n = 5$ for sham, $n = 7$ for PBS and PLN-ASO 50 mg/kg, and $n = 8$ for control ASO and PLN-ASO 25 mg/kg). Representative image of 1 out of 4 membrane stains (Supplementary Fig. 20). Ratio of SERCA2 to PLN is shown as fold change relative to PBS. **c** Individual MRI assessment and quantification of the change in left ventricular ejection fraction (LVEF), left ventricular end-systolic volume (LVESV), and left ventricular end-diastolic volume (LVEDV) between treatment initiation (6 weeks post MI) and 5 weeks after start of treatment showing ($n = 5$ for sham, $n = 7$ for PBS, $n = 8$ for control ASO, $n = 8$ for PLN-ASO 25 mg/kg, and $n = 6$ for PLN-ASO 50 mg/kg). **d** Hemodynamic assessment of the maximum/minimum first derivative of LV pressure (Max dP/dt and Min dP/dt) performed at study end, 5 weeks after treatment start, at baseline and upon increasing dobutamine doses of 0, 1, 2, and 6 µg/kg/min ($n = 5$ for sham, $n = 7$ for PBS and PLN-ASO 50 mg/kg and $n = 8$ for control ASO and PLN-ASO 25 mg/kg). **e** Left ventricular sections from all treatment groups were stained for Caveolin-3 to visualize t-tubules. The density and proportions of transverse and longitudinally-oriented elements were determined using custom-made software in MatLab (see methods for details). Number of cells/animals analyzed: sham: $n = 83$ (rats $n = 5$), PBS $n = 82$ (rats $n = 5$), control ASO $n = 76$ (rats $n = 4$), PLN-ASO 25 mg/kg $n = 35$ (rats $n = 3$), and PLN-ASO 50 mg/kg $n = 54$ (rats $n = 5$). A repeated measures model with one-sided paired contrasts was used to compare the mean differences between treatments and a control group (PBS). Dunnett's test was used to adjust $p$ values for multiple contrasts with the vehicle/reference group. For other data, one or two-way analysis of variance was used for analyses, with PBS treated animals as the reference group in multiple comparison analyses. Asterix denotes significance level based on two-sided $P$ values compared to PBS with: *$P$ value < 0.05; **$P$ value < 0.01; ***$P$ value < 0.001; ****$P$ value < 0.0001. Single values are depicted, and error bars represent standard error of the mean (SEM). ASO antisense oligonucleotide, LAD left anterior descending artery, MRI magnetic resonance imaging. Source data are provided as a Source Data file.

$-52\mu l$ ($\pm 28 \mu l$), $P$ value $= 0.0005$ in PBS versus high-dose respectively, Fig. 4c). Furthermore, cardiac contractility, as assessed by LV dP/dtmax, was reduced by the MI compared to sham, and this effect was also reversed by PLN-ASO treatment in a dose-dependent manner. Indeed, high-dose PLN-ASO treatment significantly increased dP/dtmax values compared to either PBS, control ASO, and low dose PLN-ASO groups (Fig. 4d). No differences in cardiac relaxation (LV dP/dtmin and tau, Fig. 4d and Supplementary Fig. 21) were observed between treatment groups. Upon increasing dobutamine doses, all groups responded with corresponding increases in inotropy and lusitropy, suggesting a preserved/maintained cardiac reserve (Fig. 4d and Supplementary Fig. 21). No significant effects were observed on heart rate, systemic mean arterial blood pressure, and heart weight by PLN-ASO treatment (Supplementary Figs. 21, 22). However, reductions of body weight gain over the course of the study were observed with control and PLN-ASO treatment (Supplementary Fig. 22).

Reduced cardiomyocyte contractility during HF is centrally linked to impaired $Ca^{2+}$ cycling following disruption of t-tubules[36]. Indeed, an intact t-tubule network is a prerequisite for proper excitation-contraction coupling. Since previous work has indicated that recovery from HF following SERCA overexpression critically involves recovery of t-tubule structure[37], we hypothesized that similar changes underlie recovery of systolic function in post-MI rats treated with PLN-ASO[36]. To this end, we assessed t-tubule density and organization by confocal microscopy in ventricular sections labeled with caveolin-3 (Fig. 4e). We first analyzed the density of T-tubules exhibiting a transverse orientation, as these are the primary sites of $Ca^{2+}$-induced $Ca^{2+}$ release in cardiomyocytes[36]. In agreement with previous work[38], transverse t-tubule density was observed to be decreased in the myocardium proximal to the infarction (max distance $= 500\ \mu m$) in hearts treated only with PBS. This effect was markedly attenuated in animals that received the highest ASO dose (50 mg/kg), as transversely-oriented tubules were reestablished to levels similar to those observed in sham-operated controls. Densities of longitudinal elements, which are frequently observed in diseased cardiomyocytes, tended to be increased in all post-infarction treatment groups. In summary, PLN-ASO prevented deterioration of cardiac contractility in the post-MI heart, and this

therapeutic effect was linked to normalization of t-tubule structure in cardiomyocytes.

## Discussion

The PLN/SERCA2a interaction is a well-accepted therapeutic HF target, but to date no successful pharmacological strategy has been identified. Previous efforts have used different approaches, mostly focusing on gene therapy, since small-molecule approaches to specifically target the PLN-SERCA2a interaction have been challenging. In this study, we present a strategy using ASOs to decrease *Pln* mRNA. The resulting reductions in PLN protein expression prevented or reversed cardiac dysfunction in three different rodent models of HF and cardiomyopathy, and significantly increased lifespan in a model of hereditary PLN R14del cardiomyopathy.

Our limited understanding of the mechanisms involved in the complex spectrum of HF syndromes and specifically DCM is evident by the scarcity of effective treatments. A common pathophysiological hallmark however is aberrant $Ca^{2+}$ handling in cardiomyocytes, which can include dysfunctional PLN and/or SERCA2a to a variable extent[39]. The contribution of aberrant $Ca^{2+}$ cycling and a dysfunctional PLN/SERCA2a interaction differs between diseases. Whereas PLN is likely a key disease driver in carriers of pathological PLN variants, abnormal $Ca^{2+}$ cycling is rather a consequential aggravating factor in common acquired heart disease, such as MI. In a murine transverse aortic constriction model, simulating cardiac pressure overload induced HF, Morihara et al. showed the short-term effects of a locked nucleic acid ASO targeting PLN administered via a single intravenous dose using a proprietary hydrodynamic delivery system[23]. A 6.5% improvement in FS after 1 week was observed in treated mice[23]. This minor and isolated improvement in FS was observed in a total of five mice, with no safety or efficacy data on repeated administrations, long-term follow-up or other clinically relevant models. The clinical tolerability and safety of this delivery solution and methodology is untested, as it is only used as a research tool. Conversely, we set out to evaluate a therapeutically relevant means to reduce PLN expression using cET-modified gapmer ASOs administered subcutaneously. Such ASOs, commonly referred to as generation 2.5, are currently under clinical evaluation across a number of different indications[40]. To robustly

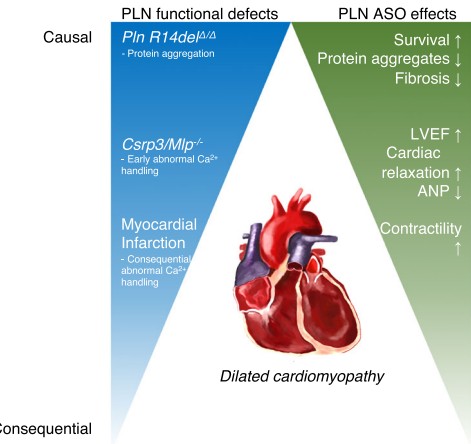

**Fig. 5 A graphical study summary showing increased beneficial PLN ASO treatment effects with increased contribution of PLN to disease.** Summary figure of study results. Three murine models of a dilated cardiomyopathy were studied, including the genetically engineered mouse models of PLN R14$^{\Delta/\Delta}$ and Csrp3/Mlp$^{-/-}$, and the rat myocardial infarction model. PLN functional defects are causative to the pathology observed in PLN R14$^{\Delta/\Delta}$, early contributing to the pathology in Csrp3/Mlp$^{-/-}$, and consequential and aggravating in myocardial infarction. In line with this, the beneficial effects of the PLN-ASO were most abundant in the PLN R14$^{\Delta/\Delta}$ and Csrp3/Mlp$^{-/-}$ models, showing almost complete prevention or reversal of disease phenotype. In our myocardial infarction model, the PLN-ASO restored cardiac dimensions and contractility, but not relaxation or left ventricular ejection fraction. ASO antisense oligonucleotide, ANP atrial natriuretic peptide.

test the efficiency and potential clinical translatability of PLN-ASO treatment in a variety of cardiac disease, covering the aforementioned aspects, we selected three different rodent HF models: two mouse DCM models, one driven by mutant PLN R14del, and the other driven by a Cspr3/Mlp$^{-/-}$ mutation, and a rat post-MI model to represent common acquired heart disease (Fig. 5. Summary figure).

Of all currently known *PLN* variants, the PLN R14del pathogenic variant is the most prevalent with a population over 1200 carriers identified to date[41]. A recent study described high similarity in features of human disease and the PLN R14$^{\Delta/\Delta}$ mouse model, which we also used in these studies[27]. This first study also included pharmacological interventions, but showed that standard HF therapy including a beta-blocker and aldosterone antagonist was ineffective in preventing cardiac dysfunction in the PLN R14$^{\Delta/\Delta}$ mouse model[27]. The more specific PLN-ASO therapy, in contrast, could prevent all characteristic signs of PLN R14del cardiomyopathy: PLN protein aggregation, severe cardiac fibrosis, cardiac dilatation, impaired contractility, and abnormal ECG readings. This was accompanied by a significantly increased lifespan by at least threefold, from the average age of 8 weeks to a minimum of 24 weeks. Towards study end, treated PLN R14$^{\Delta/\Delta}$ mice started to show signs of a cardiac functional decline which might be due to the relatively low maintenance dose of PLN-ASO. However, we could not quantify PLN protein downregulation in the PLN-ASO treated mice vs. vehicle during follow-up, since all vehicle-treated control mice died before the age of 9 weeks. Given the present strong PLN-ASO efficiency data, this treatment strategy holds promise of becoming a first precision drug for this group of patients, for whom no adequate treatment is currently available other than a heart transplant. To demonstrate the general applicability of a PLN-ASO, we corroborated our findings in two independent murine models, where HF is not primarily driven by aberrant Ca$^{2+}$ signaling.

The exact mechanism by which *Cspr3/Mlp*$^{-/-}$ results in cardiac disease is still under investigation. However, Ca$^{2+}$ handling abnormalities were shown to be present, and to contribute to the phenotype in *Csrp3/Mlp*$^{-/-}$ mice in previous in vivo studies[32,42]. Recently, engineered *Csrp3/Mlp*$^{-/-}$ induced pluripotent stem cell-derived cardiomyocytes were shown to develop a hypertrophic and HF phenotype due to abnormal Ca$^{2+}$ handling which could be improved by exposure to a L-type Ca$^{2+}$ channel blocker[43]. Although we did not observe impaired Ca$^{2+}$ fluxes in the *Csrp3/Mlp*$^{-/-}$ cardiomyocytes, the PLN-ASO improved Ca$^{2+}$ handling as hypothesized. Correspondingly, the PLN-ASO normalized cardiac function and dimensions in vivo, comparable to earlier results describing the prevention of the *Csrp3/Mlp*$^{-/-}$ cardiac phenotype when combined with the genetic deletion of *Pln* or DWORF, a SERCA2a activator, overexpression[8,32].

Aberrant Ca$^{2+}$ signaling and downregulation of SERCA2a has been established by many groups in the well-studied rat MI model. Interference with PLN/SERCA2a in MI models was reported to have some beneficial effects, mostly when used as a preventive strategy and on reducing arrhythmias[35,44,45]. Here, we aimed to intervene with the progression of LV remodeling, both at the whole-heart and subcellular level, after this process was already set in motion. Thus, we initiated PLN-ASO treatment 6 weeks post-MI, and tracked cardiac structure and function over the ensuing weeks. We observed that cardiac dimensions progressively increased over the course of the study in placebo-treated rats, and these changes were halted or even improved by PLN-ASO. In addition, impairment of cardiac contractility in post-MI hearts was significantly reversed by PLN-ASO. Cardiac relaxation was not improved, potentially highlighting the complexity of diastolic relaxation and its dependence on cardiomyocyte morphology and cardiac tissue composition alongside diastolic Ca$^{2+}$ uptake[46,47]. Interestingly, in agreement with a moderate stage of HF induced in placebo-treated rats, overall SERCA levels did not decline compared to sham control rats. These findings suggest that a supranormal SERCA/PLN ratio can be therapeutic in treating HF, even in the absence of SERCA depression and an advanced stage of the disease. Interestingly, improvement of cardiac contractility was linked to preservation of cardiomyocyte t-tubule structure. Indeed, t-tubule disruption is a well-established mechanism for impaired contractile function in HF, and pharmacological treatment of heart failure is frequently associated with t-tubule repair[36]. Our current observations appear to be in agreement with previous work that has indicated that SERCA overexpression can directly repair t-tubule derangement in this disease[37]. While the underlying mechanisms are unclear, recent work has indicated that declining SERCA expression and t-tubule structure are closely linked to the dilation of ventricular dimensions during HF progression[38,48]. The current study supports this link, and indicates that increasing SERCA activity can reverse detrimental remodeling at both the whole-heart and subcellular level. We speculate these effects could be more pronounced with an earlier timed intervention, prior the onset of extensive scarring and remodeling. Compared to gene therapy utilized in earlier studies, these promising results in rodent models of MI, in PLN R14del related cardiomyopathy and DCM (*Cspr3/Mlp*$^{-/-}$), highlight the therapeutic potential of targeting cardiac PLN[35,44,45].

Improvements in cardiac function were achieved with a range of PLN reductions spanning 30–80% in *Cspr3/Mlp*$^{-/-}$, PLN R14$^{\Delta/\Delta}$, and post-MI rats (Figs. 2, 3, 4). This would be expected given previous observations of the gene dosage effects of decreased PLN expression and increased contractility in PLN intact, heterozygous and null mice[49]. Understanding the therapeutic window of PLN inhibition remains speculative and will ultimately require careful dose titration studies in patients. Our

data demonstrate that mice with 50% PLN reductions maintain contractile reserve to β-adrenergic stimulation, and similar observations were made in post-MI rats. Due to the physiological differences in cardiac SERCA2a activity, $Ca^{2+}$ cycling and cardiac reserve in rodents vs. humans, future studies in large animal models to define the therapeutic index of PLN inhibition will be necessary before proceeding to clinical studies.

There are several limitations to our approach. As ASOs do not permanently affect gene expression, as might be achieved with gene therapy, repeated subcutaneous administrations will be required for chronic treatment. We feel this is a beneficial attribute, as repeated dosing will allow careful titration of PLN suppression to safely achieve maximally efficacious improvements in SERCA2a function. Like many traditional therapeutics, a therapeutic index can be defined with a margin of safety. This would be especially important considering the potential of the on-target safety concern of the arrhythmogenic potential of increased SR $Ca^{2+}$ leak following excessive SERCA2a activity. We do not have data on a scrambled ASO control in the PLN R14del mice study as we prioritized maximizing the group sizes for either vehicle or PLN-ASO treatments. Future studies will need to determine whether any ASO off-target effect can influence R14del aggregate formation or aggregate toxicity. The studied PLN $R14^{\Delta/\Delta}$ mouse model is homozygous, and observed effects might thus not be directly translatable to human heterozygous PLN R14del carriers. $Ca^{2+}$ fluxes were assessed using relative fluorescence signals, not allowing the absolute quantification of intracellular, or sarcoplasmic reticulum, $Ca^{2+}$ levels. Future studies will need to determine the extent of SR $Ca^{2+}$ load following PLN inhibition, and whether such effects occur similarly in rodents, large mammals and PLN $R14^{\Delta/\Delta}$.

Overall, our PLN-ASO experiments suggest potential to further develop such a therapeutic strategy as both a personalized medicine for PLN mutation carriers and as a more generalized treatment for patients with DCM or HF with reduced ejection fraction (HFrEF).

## Methods

All experimental protocols were approved by the local Animal Ethical Committee (permit numbers: AVD10500201583, IVD1583-02-001 and IVD1583-02-006 for PLN-R14del mice, approved by the Experimental Animal Committee of the University Medical Center Groningen, the Netherlands, 86-2015 for rat MI study, 43-15 for Cspr3/Mlp$^{-/-}$ study 1 and 2 approved by the Jordbruks verket—regional animal testing ethical board in Sweden and P-0308-100113 approved by Ionis Institutional Animal Care and Use Committee in the USA and were performed conform the ARRIVE guidelines[50].

**Antisense oligonucleotides**. ASOs used in this report were 16mer (S)-constrained ethyl (cEt) modified 3-10-3 gapmers containing three cEt-modified ribonucleotides in each end and 2′-deoxynucleotides in the middle portion of the molecule with a phosphorothioate backbone as previously reviewed[51]. For palmitic acid conjugated ASOs, palmitate was conjugated at the 5′-terminus of the ASO with a phosphodiester linkage as previously described[21]. The nucleotide sequence of ASOs used in the cardiomyocyte and HF studies are shown in the Table 1.

**Adult mouse cardiomyocyte isolation**. Adult cardiomyocytes were isolated using a Langendorff apparatus (Harvard Apparatus). Isolated hearts were kept in ice-cold isolation buffer (NaCl: 12 mM, KCl: 0.54 mM, HEPES: 2.5 mM, $MgCl_2$-$_6H2O$:

0.05 mM, $NaH_2PO_4$: 0.04 mM, $C_6H_{12}O_6$: 24.20 mM) prior to cannulation and perfusion. Mouse hearts were cannulated through the aorta and perfused with isolation buffer for 2–4 min until there was no remaining blood. Next, the hearts were perfused with liberase solution (30 µg/ml) dissolved in isolation buffer containing $CaCl_2$-$2H_2O$:12.5 µM for 5–7 min. Immediately after, liberase perfusion, the left and the right atria were removed, and the heart were cut into small pieces (2 × 2 mm). Using small forceps, the pieces of muscle were gently agitated to further dissociate the tissue, until it had mostly dissolved. Using a disposable plastic transfer pipette, the solution was then gently pipetted to achieve a homogenous solution. The homogenized solution was transferred to a falcon tube and passed through a 250 µm filter to remove larger tissue debris. Cardiomyocytes were left at room temperature for 15 min to sediment. Once cardiomyocytes settled, supernatant was removed, and cells were resuspended in fresh isolation buffer.

**Neonatal mouse cardiomyocyte isolation and culture**. Neonatal mouse cardiomyocytes were isolated from the ventricles of 3-day-old mice homozygous for PLN R14del or wild-type C57Bl6/J littermates after decapitation. After trypsinization, cells were passed through a 40µm cell strainer and plated for 70 min to get rid of large debris and fibroblasts. Cells were plated on culture plates with microcontact printed flexible PDMS substrates at 50.000 cells per well and cultured at 37 °C under 5% $CO_2$. Media for the first 24 h was MEM supplemented with 5% FCS and 1% penicillin–streptomycin, hereafter DMEM with 10% FCS and 1% penicillin–streptomycin was used. Media was additionally supplemented with 1% BrdU for the first 4 days. Cells where treated with either 15 µM PLN-ASO or PBS for 30 h. After an additional 24 h, cells were incubated with 2.5 µM Fluo-4 (F14201, Invitrogen) for 20 min at RT, followed three washes with warm PBS and placed in RPMI media with HEPES 1:1000. Cell were then paced at 3 Hz and live imaging was recorded at 74 fps using a DeltaVision Microscope System from GE Healthcare[52].

**Intracellular $Ca^{2+}$ transients imaging and analysis**. Adult mouse cardiomyocytes were incubated with Fluo-4 (Molecular Probes) at 5 µM for 30 min. Fluo-4 was incubated in Tyrode's solution for 30 min to wash and allow for Fluo-4 deesterification before imaging. For $Ca^{2+}$ recordings, cells were place in Tyrode's solution containing: $CaCl_2$: 180 mM, NaCl:130 mM, KCl:5.4 mM, HEPES:25 mM, $MgCl_2$·$6H_2O$: 0.5 mM, $NaH_2PO_4$: 0.4 mM; pH 7, 4 at 37 °C. Cardiomyocytes were paced at 1 Hz and live imaging was recorded at 100fps. Loaded cardiomyocytes were mounted on glass bottom organ dishes (Matek, P35G-1.5-14-c) coated with matrigel (Gibco, 356230). Cell were imaged using a Nikon Eclipse with a Yokogawa CSU-X1 spinning disk and an EM-CCD Andor camera for high speed acquisition. Individual whole cell $Ca^{2+}$ transient intensities were analyzed in R software. $Ca^{2+}$ transients are calculated as $\Delta F/F_0$, with signal levelling and background normalization, where $F_0$ is background fluorescence and F if fluorescence intensity. For each cell, Amplitude (peak intensity), Time to Peak, Time to Base, Upstroke Velocity and Decay Velocity were calculated. Isoproterenol was used at a final concentration of 250 nM.

### In vivo studies

*Treatment of PLN $R14^{\Delta/\Delta}$*. Generation of PLN-$R14^{\Delta/\Delta}$ mice was previously described[27]. In short, the third, and coding, exon of the *Pln* gene was flanked by *loxP* sites (*floxed*) and followed by a third exon of the *Pln* gene with the c.40-42del AGA mutation in a C57BL6/N mouse line. Mice were crossed with germline *Cre*-expressing mice, replacing the wild-type PLN exon-3 with the PLN-R14del exon-3. The offspring were backcrossed into a C57BL6/J background for at least five generations. ASOs were formulated in PBS and subcutaneously injected into the neck compartment at a dose of 100 mg/kg on days 0, 7, 14, and 21 followed by 50 mg/kg on days 49, 77, 105, and 133 (see Fig. 3a for dosing scheme). Mice were randomly allocated to groups based on gender. For all PLN $R14^{\Delta/\Delta}$ experiments, wild-type data for reference were, in accordance with the 3 R's (replacement, reduction and refinement) reused from a previous report in case of cardiac MRI and ECG experiments[27]. For RNA, protein and histological assessments, new experiments were performed on previously acquired material. The investigators were blinded to experimental settings during in vivo experiments, data acquisition and analysis. Equal ratios of male and female mice were used for all experiments. Mice were housed on a 12 h light/12 h dark cycle with *ad libitum* access to chow and water, ambient temperature at 20–24 °C and 45–65% humidity. Imaging and euthanasia were performed under 2–3% isoflurane (TEVA Pharmachemie, The Netherlands) anesthesia mixed with oxygen via an aerial dispenser. Heart and respiration rate and body temperature were continuously monitored.

*Cardiac MRI of PLN $R14^{\Delta/\Delta}$ mice*. Mice were anesthetized with isoflurane (2%) and imaged in a vertical 9.4-T, 89-mm bore size magnet equipped with 1500 mT/m gradients and connected to an advanced 400 MR system (Bruker Biospin) using a quadrature-driven birdcage coil with an inner diameter of 3 cm. Respiration and ECG were continuously monitored and maintained at 20–60 breaths per minute and 400–600 bpm, respectively. ParaVison 4.0 and IntraGate software (Bruker Biospin GmH) were used for cine MR acquisition and reconstruction. After orthogonal scout imaging, short-axis (oriented perpendicular to the septum) cardiac cine MR images were acquired. Semi-automatic contour detection software

| Table 1 Antisense oligonucleotide sequences. | |
| --- | --- |
| **ASO** | **Sequence (5′–3′)** |
| ASO#26 | GATAAATGTACTCTGT |
| ASO#27 | GCATATCAATTTCCTG |
| ASO#136 | GTAACTTATATCTTGG |
| Control-ASO | GGCCAATACGCCGTCA |
| The sequences of the antisense oligonucleotides used for the in vivo experiments. | |

(CVI[42], version 5.6.6, Circle Cardiovascular Imaging, Canada) was used for the determination of the LV end-diastolic volume, LV end-systolic volume, stroke volume, and ejection fraction (EF), as described[53].

*Electrocardiography of PLN R14$^{\Delta/\Delta}$ mice.* ECG recordings were acquired using two-lead subdermal needle electrodes, connected to a PowerLab 8/30 data acquisition device (model ML870, ADInstruments, Australia) and an animal Bio Amp biological potential amplifier (model ML136, ADInstruments) as previously reported[54]. RR-, PR-, QRS- and QT-intervals, P-duration, P-, Q-, R-, S- and T-amplitudes, ST-height and heart rate were analysed using the ECG Analysis module in the Lab-Chart Pro software version 8 (ADInstruments).

*Cspr3/MlpKO mice maintenance, and study details.* C57BL6/N *Cspr3/Mlp$^{-/-}$* males and females were used for several in vivo studies, see Supplementary Data 1 for experimental details. Mice were housed on a 12 h light/12 h dark cycle with *ad libitum* access to chow and water, ambient temperature at 21–22 °C and 50% humidity. ASOs were formulated in PBS and subcutaneously injected into the flank or neck compartment at a dose of 100 mg/kg on Day 0 and 50 mg/kg on Days 7 and 14 for palmitate conjugated PLN-ASO, and at a dose of 100 mg/kg on Days 0, 7, 14, and 21 for parent PLN-ASO (see Fig. 3a for dosing scheme). The mice were randomly allocated to groups based on body weight, gender, age, and LVEF. For echocardiography measurements, animals with LVEF < 45% were included in analysis. For hemodynamic and RT-PCR analysis, the entry criteria for mice were high plasma ANP levels (>119 ng/ml), followed by randomization into two treatment groups based on EF% baseline levels (Supplemetary Table 1).

*Echocardiography of Cspr3/MlpKO mice.* Mice were anaesthetized using 2.5% isoflurane (Forene®) mixed to air and then maintained on 1.5–2% isoflurane during the assessment. During the examination the mice were kept on a Physio Plate (Visualsonics, Canada) to keep normal body temperature and to monitor ECG and respiration. The ultrasound probe (MS400, 18–38 MHz) was connected to an ultrasound biomicroscope (Vevo 2100 System, Visualsonics, Canada). An LV long-axis parasternal B-mode in long-axis view was captured, followed by a 90° clockwise rotation of the ultrasound probe adjusted to the level just caudal to the mitral level to obtain short-axis B-MODE sequences and motion-mode (M-MODE). Anterior and posterior wall thickness as well as end-diastolic LV dimension was measured using the American Society of Echocardiography leading edge. End-diastole was defined at the R-tag in the ECG signal. LV wall thickness was calculated as the average of the anterior and the posterior wall thickness. LVM was calculated based on the uncorrected cubic formula: LVM = l.055 × [(P + A + EDD)$^3$ − (EDD)$^3$], where P and A indicate Posterior and anterior wall thicknesses, respectively, and EDD is the end-diastolic LV diameter. FS and EF were calculated from short-axis MMODE measurements as follows: FS (%) = 100 × [(LVIDd − LVIDs)/LVIDd; EF(%) = 100 × [(LVIDd$^3$ − LVIDs$^3$)]/LVIDd$^3$; CO(mL/min) = EF × LVIDd$^3$ × heart rate/100. LVIDd and LVIDs are LV internal dimensions (LVID) during diastole and systole, respectively. All measurements were averaged calculated from three consecutive cardiac cycles with animal's treatment group blinded to the sonographer. The animals were euthanized following echocardiographic examination. The hearts were punctured for a terminal blood sample using 1 mL syringes Omnifix-F BRAUN and TERUMO NEOLUS 23G × 1". Whole blood samples were mixed with the EDTA anticoagulant and stored at 2 to 8 °C. For plasma samples a microvette 500 LH Li-Hep for total protein, albumin, haptoglobin, alpha1 glycoprotein was used and samples were frozen at −80 °C until analysis. Heart weight and tibia length were measured. For histologic examination, organs were fixed in 10% buffered formalin, followed by paraffin embedding. The investigators were blinded to experimental settings during all data acquisition and analysis.

*Hemodynamic measurements of Cspr3/MlpKO mice.* Mice were anesthetized with a mixture of ketamine (100 mg/kg) and xylazine (5 mg/kg) intraperitoneally. After endotracheal intubation, the mice were connected to a volume-cycled rodent ventilator. A PE-50 catheter was placed in the left jugular vein and used for intravenous access. Through the right carotid artery, a 1.8-French high-fidelity catheter tip micromanometer was inserted via a small incision, and the tip was manipulated across the aortic valve into the left ventricle (LV). A bilateral vagotomy was performed. When LV pressure and heart rate became stable, dobutamine was given intravenously by an infusion pump, at a rate of 0, 0.75, 2, 4, 6, and 8 µg/kg/min. When a stable and reproducible pressure reading was obtained, LV pressure, Tau, Maximum positive and negative first derivative of left ventricular pressure (±dP/dtmax) were recorded and averaged from 12 beats. The body temperature was maintained at 37 °C by external heating both from the table and from a heating lamp during the experiment.

*Rat myocardial infarction and treatment.* Male Lewis rats at 8 weeks of age and a weight of ~250 g were housed on a 12 h light/12 h dark cycle with *ad libitum* access to chow (R3 (Lactamin®)) and water, ambient temperature at 21–22 °C and 50% humidity. For induction of permanent LAD occlusion, the animals were anaesthetized with Isofluran®, intubated, and connected to a rat ventilator. The rats were ventilated with air ~1100 ml/min and oxygen ~100 ml/min (~50) strokes/min, (~4 µl tidal volume). Core temperature was maintained at 37.5 ± 1 °C by a heated

operating table and a heating lamp controlled by a rectal thermometer. Electrodes were inserted under the skin to register ECG monitored and recorded (PharmLab). The rats were subjected to left thoracotomy at the fifth intercostal space ~2 to 3 mm to the left of the sternum. A rib spreader was used to keep the incision open. The pericardium was then opened and a ligature with a 7–0 suture was placed around the left coronary artery and the artery was occluded by tying the ligature. Induction of ischemia was confirmed by observed paleness of the heart distal from the suture and ST-elevation of the ECG signal. The chest was then sutured, and the rat was monitored during continued maintenance of body temperature and ventilation until it regained consciousness and could be disconnected. Six weeks post LAD occlusion rats were randomized into four treatment groups (PBS, Control-ASO, PLN ASO 25 mg/kg, PLN ASO 50 mg/kg) and treatment was initiated. Randomization was based on plasma cTnI levels one day post-infarction (LVEF% to cTnI correlation was used to estimate LVEF%), as well as visual infarct size score[55], BW, and HR at baseline (6 weeks post MI). Parent rat PLN-ASO and Control ASO were formulated in PBS, and were subcutaneously injected into the neck compartment at a dose of 50 mg/kg (PLN-ASO, Control-ASO) or 25 mg/kg (PLN-ASO) twice weekly at week 6 and 7, and once weekly at week 8–11 of study (see Fig. 3a for dosing scheme).

*Magnetic resonance imaging of rats.* All MRI scanning was performed on a 4.7-T Bruker Biospec (Bruker BioSpin GmbH, Germany) using a 72-mm quadrature coil (Rapid Biomedical GmbH, Germany). Animals were placed in the supine position and exposed to inhalation anesthesia (Isoba® vet isoflurane, Schering-Plough Ltd., England). Physiological monitoring included temperature assessments with a rectal probe, respiration rate and ECG evaluation via the SA instruments platform (SA Instruments, Inc., USA). Animals temperature was maintained at 37.0 ± 0.2 °C via circulating perfluorocarbon and respiration rate was kept at 60 ± 5 breaths per minute. Twelve short-axes CINE time series (temporal resolution 8 ms) covering the left ventricle and two perpendicular long-axis slices were acquired for the assessment of systolic function using a gated gradient-echo imaging sequence. The other sequence parameters were a 2.1-ms echo time, 25° flip angle, 128 × 192 image matrix, 200-kHz/pixel bandwidth, 1.5 mm slice thickness, and 40 × 60 mm² field-of-view. The left ventricles were manually delineated by an expert, and LVEF, LVESV, and LVEDV were derived via the post-processing software package Segment (Medviso AB, Sweden).

*Hemodynamic measurements of rats.* At study week 10–11, the animals were anaesthetized with isoflurane (Attane vet, VM Pharma AB, Stockholm, Sweden) gas (5%) in a gas chamber and kept anaesthetized by breathing isoflurane gas through a mask. A catheter (plastic tubing PE 10 connected to a PE 50, Intramedic® polyethylene tubing, Becton Dickinson, Sparks, MD, USA) was inserted in the right femoral artery for registration of mean arterial blood pressure (MAP) and heart rate (HR). The arterial catheter was flushed with 10 µL/min saline (9 mg/mL, Fresenius Kabi AG, Bad Homburg, Germany) throughout the experiment to maintain patency. An electrocardiogram (ECG) was recorded from skin electrodes. Signals from ECG, MAP, and HR were recorded and sampled by using a computer and software (PharmLab V6.6, AstraZeneca R&D Mölndal, Sweden. Ventricular function was measured via a pressure catheter (1.4F, SPR-847, Millar Instruments, Houston, Texas, US), inserted into the left ventricle via the right carotid artery and dP/dt max/min, ESP, EDP and Tau were measured with and without dobutamine infusion (1, 2, 6 µg/kg × min). The body temperature was maintained at 37 °C by external heating both from the table and from a heating lamp during the experiment.

**T-tubule imaging and analysis.** Myocardial rat sections were first de-paraffinized (heating at 60 °C for 60 min) and then submerged in Histo Clear Tissue Clearing Agent (National Diagnostics) for 2 × 3 min. Thereafter, sections were sequentially submerged in 99.9% ETOH and 96% ETOH, and then rinsed three times in dH20. Antigen retrieval was performed by submerging sections in boiling sodium citrate buffer (10 mmol/l C6H5O7Na32H2O, 0.05% Tween20) and steaming using a vegetable steamer (Philips DampfgarerHD9140) for 30 min. After sections had cooled down, they were washed in PBS before permeabilization (0.1% Triton X-100). The sections were then washed with PBS, and blocked with Image-iTTMFX Signal Enhancer (ThermoFisher). Samples were incubated with Caveolin-3 primary antibody (ab2912, Abcam; 1:50 dilution) overnight, and washed with PBS before incubation with Alexa Fluor 488 goat anti rabbit (ab150077, Abcam; 1:100 dilution). Sections were imaged on an LSM800 scanning confocal microscope (Zeiss GmbH, Jena, Germany) using a ×63 objective. Confocal image processing and quantification of t-tubule structure. All images were deconvolved using Huygens Essential software (Scientific Volume Imaging, Hilversum, Netherlands). Using custom-made software in MatLab (The Mathworks, Natick, MA, USA), t-tubule density was determined by first thresholding mean image fluorescence intensity of the entire cell. T-tubule density was then calculated for the cell interior, defined as the above-threshold area divided by the cross-sectional area of the entire cell[56]. T-tubule organization was analyzed by first removing the sarcolemma from the thresholded images before skeletonization. The proportions of transverse- and longitudinally-oriented tubules were then analyzed using a custom macro in MatLab (see for details[38]).

**qPCR**. Total RNA from cells and tissues were isolated using TRIzol Reagent (Thermo Fisher Scientific). cDNA was constructed using the QuantiTect RT kit (Qiagen). Relative gene expression was determined by quantitative real-time PCR (qRT-PCR) on a BioRad CFX384 real-time system using Absolute qPCR SYBR Green mix (Thermo Fisher Scientific) or Taqman assay. Gene expression levels were corrected for reference gene expression (36B4), and relative ratios compared to the experimental control group are expressed. The primers used are listed in Supplementary Data 2.

**Western blot**. Cells and tissues were homogenized in ice-cold RIPA (50 mM Tris pH 8.0, 1% nonidet P40 0.5% deoxycholate, 0.1% SDS, 150 mM NaCl or ThermoFisher, 89900) or urea containing phosphatase inhibitor cocktail 3 (Sigma-Aldrich) and protease inhibitor (Roche Diagnostics or ThermoFisher, 78440) and 15 mM NaVanadate in selected experiments. Protein concentrations were determined with a DC protein assay kit (BioRad) or BCA protein Assay Kit (Pierce, 23225). Equal amounts of protein were loaded on 10% polyacrylamide gels or a Mini Protein precast gel (Biorad, 4561085). After electrophoresis, the gels were blotted onto PVDF or nitrocellulose membranes. Membranes were then blocked and incubated overnight at 4 °C with the primary antibody, followed by 1 h incubation at room temperature with an HRP secondary antibody. Detection was performed by ECL and analyzed using ImageJ version 2.0.0. Protein quantifications were normalized to total protein staining (Revert 700) or GAPDH protein expression levels. Primary antibodies used: anti-PLN monoclonal (clone 2D12) (ThermoFisher: #MA3-922 1:1000), anti-SERCA2 (ThermoFisher: #MA3-919 1:1000 or Abcam ab150435 1:1000–1:5000), phospho-PLN (Cell Signaling: #8496 1:1000), GAPDH-HRP (1:2000, 1:35.000, Invitrogen MAS-15738).

**Immunohistochemistry**. Tissue slices were formalin-fixed, dehydrated, embedded in paraffin (Klinipath) and cut into 4-µm thick transversal sections for histological analysis. Masson's trichrome stain was performed to detect collagen deposition as a measurement of fibrosis as previously described[57]. Immunofluorescent staining for PLN was performed using a mouse monoclonal anti-PLN antibody (clone 2D12, #MA3-922, Invitrogen) (1:200) labeled with Alexa Fluor 555 (red) using an APEX antibody labeling kit (Invitrogen) according to the manufacturer's protocol. Sections were co-stained for cardiac troponin I (Abcam 47003 1:100) and DAPI (Vector Laboratories, CA, USA) to stain nuclei blue.

**Plasma measurements**. For the Pln R14del studies, Troponin I was quantified using a Sandwich ELISA (LSBio, LS-F24180) in 10× diluted plasma according to manufacturer protocol. AST and ALT were quantified in plasma diluted 4× with NaCl 0.9% according to IFCC with pyridoxal phosphate activation, on a Cobas 6000 (c502) chemistry platform (Roche, Mannheim, Germany). For the other studies, AST and ALT were measured on a clinical analyzer (Olympus, Center Valley, PA). ANP was measured following plasma extraction via a competitive ELISA, as described in Catalog # EIAANP (ThermoFisher, CA)

**RNA sequencing**. RNA was extracted from pulverized LV tissue using Lysing matrix D beads (MP Biomedicals) and standard TRIzol extraction (ThermoFisher scientific). RNA quality was determined using RNA Pico Chips on Bioanalyzer 2100 (Agilent) and Truseq stranded mRNA libraries (Illumina) were generated from high quality total RNA (RNA integrity number >8.0). Samples were subjected to single end sequencing on Next Seq 500 platform (Illumina). Reads were aligned to mouse reference genome (mm10) using STAR 2.4.2a[58] and read count analysis was performed using htseq-count 0.6.1[59]. Differential expression analysis was performed using DeSeq2 1.24.0[60]. Gene set enrichment analysis was performed using Fgsea 1.10.0[61] and MsigDB v6.2[62]. Genes were considered differentially expressed at a false discovery rate <0.05. Top 200 genes determining the principal components 1 and 2 were assessed for the pathway analyses. Pathways were analyzed using g:Profiler and the Kyoto Encyclopedia of Genes and Genomes database.

**Statistics**. Distinct samples were used for all analyses. All data are represented as means ± standard error of the mean (SEM). Two sample $t$ test were used to compare two groups with normal distributions, and Wilcoxon rank-sum for 2 group comparisons when non-normally distributed. In case of more than two group comparisons, one-way or two-way ANOVA with post hoc Bonferroni tests were utilized, unless mentioned otherwise. All analyses were carried out using GraphPad Prism software version 8.02 (GraphPad Software Inc.) or STATA version SE 16.0 (StataCorp). All reported $P$ values are two-sided unless otherwise noted, a two-sided $P$ value of <0.05 was considered statistically significant. All exact $P$ values are listed in Supplementary Data 3.

**Reporting summary**. Further information on research design is available in the Nature Research Reporting Summary linked to this article.

## Data availability
The data supporting the findings from this study are available in the article and the Supplementary Information. RNA-sequencing data are available in the GEO under accession number GSE151156 at. Any remaining raw data will be available from the corresponding author upon reasonable request. Source data are provided with this paper.

## Code availability
The codes for the RNA-sequencing analyses are available as Supplementary Data 4, 5 and the codes for the T-tubule analysis in MatLab are available at: https://gitlab.com/louch-group/t-tubules-script.

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

## Acknowledgements

The excellent technical assistance of Martin Dokter, Silke Oberdorf-Maass, Kees van der Kolk, Marloes Schouten, and David Kijlstra are gratefully acknowledged. We thank Utrecht Sequencing Facility (USEQ) for providing sequencing service and data. We acknowledge Peter Konings for the statistical analyses of experiment related to the Cspr3/Mlp$^{-/-}$ mouse studies 1 and 2, and the adult cardiomyocyte calcium imaging, as well as the rat MI study. Additionally, Kirk Peterson, Nancy Dalton, and Yusu Gu of the Seaweed Canyon Cardiovascular Physiology Laboratory, University of California at San Diego School of Medicine are acknowledged for their excellent technical assistance. H.H.W.S., T.R.E. and R.A.B. received support from the de Boer Foundation, Ubbo Emmius Foundation, PLN Foundation, the Netherlands Heart Foundation (CVON DOSIS, grant 2014–40 and CVON PREDICT2, grant 2018–30) and the leDucq Foundation (Cure PhosphoLambaN induced Cardiomyopathy (Cure-PLaN). R.A.B. receives further support from Netherlands Heart Foundation (CVON SHE-PREDICTS-HF, grant 2017–21; CVON RED-CVD, grant 2017–11) and the European Research Council (ERC CoG 818715, SECRETE-HF). Utrecht Sequencing Facility is subsidized by the University Medical Center Utrecht, Hubrecht Institute and Utrecht University. C.J.B. received support from the European Union's Horizon 2020 research and innovation program under the Marie Skłodowska-Curie grant agreement No. 751988. R.K. received funding from AstraZeneca, Karolinska Institutet, Leducq Transatlantic Network of Excellence #13CVD04, Hjart och Lungfonden #20200265, and German Research Foundation (DFG) #Kn448/9-1 and 10-1. K.R.C. received funding from the ERC (ERC grant No. 743225) and the Swedish Research Council.

## Author contributions

N.G.B.: designed and conducted the PLN R14del studies, acquired funding, analysed and interpreted the data and wrote the paper. D.S.: designed and supervised Cspr3/Mlp$^{-/-}$ and rat MI studies, designed PLN R14del study, performed rat WB, acquired funding, interpreted data, and wrote the paper. S.T.Y.: performed and analysed studies to identify PLN ASO leads and Cspr3/Mlp$^{-/-}$ pharmacology studies. A.H.: performed, analysed, and interpreted WB analysis of Cspr3/Mlp$^{-/-}$ study #1 + 2, and Calcium imaging on Cspr3/Mlp$^{-/-}$ cardiomyocytes, IHC on Cspr3/Mlp$^{-/-}$ hearts. H.H.W.S.: designed the PLN R14del mouse model, acquired funding and interpreted data. Z.E.: planned and performed part of the Cspr3/Mlp$^{-/-}$ study #1 + 2 and the adult cardiomyocyte isolation, and performed and analyzed the protein extraction of rat MI organ samples. Humam Siga: performed and analysed echocardiography measurements of Cspr3/Mlp$^{-/-}$ study #2, assisted with organ sampling of Cspr3/Mlp$^{-/-}$ study #1 + 2, M.P.: designed,

performed and analyzed echocardiography of *Cspr3/Mlp*$^{-/-}$ study #1, and performed rat MI surgery. T.A.: designed, and performed rat MI surgery. M.Z.: designed, performed, and analyzed MRI measurements for rat MI study. S.P.: designed, performed, and analysed hemodynamic measurements for rat MI study. T.R.E.: assisted in part of the PLN R14del studies and analysed and interpreted data. N.B.: assisted in part of the PLN R14del studies, interpreted data. M.F.H.: performed experiments and interpreted data. C.J.B.: performed RNA-sequencing experiments, analyzed, and interpreted data. M.F.: performed T-tubule analysis, analyzed, and interpreted data. E.v.R.: contributed to the experimental design of the RNA-sequencing experiments and interpreted the data. S.D.: performed RNA-sequencing experiments, analyzed, and interpreted data. W.E.L.: contributed to the experimental design of the t-tubule analysis and post-MI rat study, interpreted data. Q.D.W.: contributed to experimental design and data interpretation of *Cspr3/Mlp*$^{-/-}$ in vivo studies and rat MI study. K.R.C.: contributed to experimental design and data interpretation of all in vivo studies. R.F.D.: contributed to experimental design and data interpretation of *Cspr3/Mlp*$^{-/-}$ in vivo studies and rat MI study, secured funding. K.H.: designed PLN R14del, *Cspr3/Mlp*$^{-/-}$, and rat MI study, acquired funding, and interpreted data. R.K.: designed and supervised *Cspr3/Mlp*$^{-/-}$ studies and interpreted data. A.E.M.: designed and supervised PLN ASO lead identification, *Cspr3/Mlp*$^{-/-}$ studies, designed PLN R14del study, acquired funding, interpreted data, and wrote the paper. R.B.: designed the PLN R14del mouse model, acquired funding, and interpreted data. P.v.d.M.: designed and supervised the PLN R14del studies, acquired funding, interpreted the data, and wrote the paper. All authors discussed the results and critically revised the paper.

## Competing interests

A.H., Z.E., H.S., C.J.B., M.F., E.R. and W.E.L. declare no competing interest. D.S., R.K., A.H., M.Z., M.P., S.P., T.A., Q.W., R.F. and K.H. are employees of AstraZeneca. S.Y., S.D. and A.E.M. are employees of Ionis Pharmaceuticals. N.G.B., T.R.E., H.S., N.B., M.F.H., R.A.B. and P.v.d.M. are employees of the UMCG which received research grants and/or fees from AstraZeneca, Abbott, Bristol-Myers Squibb, Novartis, Novo Nordisk, and Roche. K.R.C. is a member of the Scientific Advisory Board and receives research support from Astrazeneca, and is a Co-Founder and Equity holder in Moderna Therapeutics. R.d.B. received speaker fees from Abbott, AstraZeneca, Novartis, and Roche.
