## [Peer Review File · Nature Communications]

REVIEWER COMMENTS

Reviewer #1 (Remarks to the Author):

The present manuscript is aimed to study the therapeutic use of antisense oligonucleotides (ASOs) targeting the expression of phospholamban (PLN) in cardiac diseases. The authors comprehensively addressed the effects of ASOs treatment in vitro as well as in vivo on three murine models of heart diseases, including two independent genetic models of dilated cardiomyopathy (DCM) and one of myocardial infarction (MI). Many ASOs were initially screened in vitro to evaluate the most efficient to downregulating the PLN expression and then, in vivo experiments were performed to select the ones with the least toxicity. Two PLN-ASOs were used in mice.

The overall work is well organized, compels a great amount of novel data, results very well presented and the methodology used is appropriate. The article is well referenced, and the conclusions are supported by the results.

Specific Concerns:

1. Although the studies in both models of DCM include intracellular calcium measurement either in neonatal or adult cardiomyocytes, it is important to study how the PLN-ASO treatment affects the calcium handling in the model of an acquired cardiomyopathy in this study, MI in rats. So, calcium measurements are recommended, especially in this model. Accordingly, the section of the study that addresses MI is evidently less exhaustively explored than the sections of DCM. For instance, SERCA2 expression (which is expected to be reduced in a HF model) is not showed. It would be very interesting to see, in this particular model, how the ratio SERCA2/PLN is affected by the treatment and its correlation with calcium decay rate, etc.
2. In line with the previous observations, the authors mention that the same screening process was carried out for selecting the most appropriate ASO for treating rats, however no data is shown. Moreover, they do not provide information about what particular PLN-ASO was administered to the rats. Please complete this missing information.
3. Why the ASO control was not used (or showed if experiments were done) in the mouse models of DCM?

Minor comments

1. Figure 2h, the ECG trace for the PLN R14 Δ/Δ +PBS is not representative of the averaged values for waves R and S, and exaggerates the differences with the other groups.
2. Figure 3e, all 4 subpanels are repeated in the Suppl. Figure 13. Please, be sure to use the supplemental figures to show only supporting data and not duplicate them.
3. Figure 4 is full of panels which could be better compiled in a table or a lesser number of panels, and move most of them to the supplemental figures.

4. Supplemental Figure 15a, the color code of the calcium transient traces seems to be switched with respect to the rest of the figure/s. It would be easier for the reader if the colors of the traces match with the ones in the bars.

5. In Supplemental Figure 15e and f the authors show APD50 and APD75, respectively. APD is not defined, but instinctively one could guess Action Potential Duration. However, the authors have not described any electrophysiology experiment measuring membrane potential of action potential. Moreover, the legend for the Suppl. Fig. 15 says: ADP50 and ADP75. Could you please clarify what parameter is depicted here?

Reviewer #2 (Remarks to the Author):

This manuscript describes the activity of antisense oligonucleotides in cardiac tissue. Relatively little good work has been done investigating the activity of ASOs in the heart, making this a novel topic.

Unfortunately, there are two big drawbacks that cannot be overlooked.

1) There is no detailed description of the assays using ASOs nor any information about their composition. The exact molecular structure of the compounds needs to be disclosed.

2) PBS is not an adequate control. For studies like these control oligonucleotides of similar sequence but with disrupted base-pairing must be used.

Point 1) can be easily corrected. Unfortunately, while 2) could have been easily addressed when these experiments were done, it would be necessary to now redo the experiments with proper controls. Sadly, because activity in the heart would be novel, I do not believe that the lack of accepted rigor can be excused in this case.

I appreciate the heart work that went into this study, but the flawed experimental design reduces my enthusiasm for the work.

Reviewer #3 (Remarks to the Author):

This study describes the rescue of three heart failure models by antisense phospholamban oligonucleotides (PLN-ASO). Two of the models were genetic mouse models of HF and the third one was an MI rat model of HF. Previous studies have supported the benefits of targeting the impaired Ca-cycling in heart failure through increasing either the SERCA2a levels or decreasing the inhibitory levels/activity of PLN. Initially 312 ASOs were evaluated by the authors, 21 were selected for further evaluation in healthy mice (3 subcutaneous administrations) and finally two ASOs were selected for evaluation in the HF models. Comprehensive studies were performed in each model using integrative approaches to evaluate the benefits of PLN-ASO. The first model is the PLN-R14del mouse and PLN-ASO prevented PLN aggregates, cardiac dysfunction and extended the life span of the animals. The second model is the Cspr3/MLP-KO mouse and PLN-ASO reversed the HF phenotype. The third model is a rat MI model where PLN-ASO improved contractility.

Concerns

Overall, the benefits associated with PLN-ASO are only observed for a very short time after the final dose administration, making the therapeutic potential of this approach rather limiting. Several ASO doses are administered in each of the models and studies are mainly performed within a week after the final dose administration. Longer term effects are not assessed except for the survival study in the PLN-R14del model (but PLN-ASO had to be administered through out the life span of the model).

The PLN-R14del model used in this study is homozygous for the R14del-PLN mutation but the patients are heterozygotes. This model shows a severe phenotype by 8-9 weeks of age. It would be more appropriate to carry out this study with PLN-ASO in the heterozygous model with the same genotype as the R14del-patients.

Fig 1e: The PLN mRNA remains low only up to Day 14 following the 3 doses. It increases after day 14 post the third injection/dose. Indeed, the PLN protein (Suppl Fig 1a) shows increases up to 50% of PBS levels by day 42. Thus, the decreases in PLN mRNA or protein are temporary and maintenance of PLN-ASO administration would be a requirement in clinical studies. Please address.

Fig 1f: neonatal myocytes are used but these studies should be performed in adult or mature cardiomyocytes. Is SR Ca-load altered by PLN-ASO? Why isn't there an increase in FS when the Ca-peak was increased? Are there any alterations in myofilament activity?

In the R14del-model: there are 4 doses at weeks 3-6 and then assessment is done a week later (at 7 weeks of age). What are the specific Ca-defects that PLN-ASO targets and restores in this model?

The R14del-PLN model shows 35% of mRNA levels compared to WT and 40% of PLN protein levels compared to WT (PBS data). Thus, if PLN is decreased, why does one need to further decrease it by PLN-ASO? If the low levels of PLN protein are due to its presence in aggregates, then one would not expect to observe similar decreases in PLN mRNA levels in this model (PBS data: Fig 2b).

Supplemental Fig 3 shows a full blot. Which bands are used to quantitate PLN in Fig 2C?

How is the PLN level quantitated in Suppl Fig 5? It is expressed as fold change (To PBS) but this is not clear. What does the PLN data in “Dissolvable in RIPA” vs “Dissolvable in UREA” mean? Which of these data in Suppl Fig 5 correspond to the data in Fig 2c?

Fig 2d: Please use arrows to point to the aggregates.

The mouse survival study required a lower maintenance dose every 4 weeks. In this follow-up, the PLN mRNA levels increased at T15-23 compared to PLN-ASO (Suppl Fig 10a) but the PLN protein levels remained low through T23 (Supl Fig 10b). How do the authors explain this apparent discrepancy between mRNA and protein levels?

Also, in Supplemental Fig 10:

- a) Although PLN-ASO was administered every 4 weeks, fibrosis increased starting at T19. At T23, fibrosis was similar to PBS: why?
- b) LVEF and LVESV showed deterioration starting at T19 and at T23, they were similar to PBS. Please discuss.
- c) Similar concerns exist with the ECG: R, S and R+S approached PBS levels by T19 and T23. Please address.

Figure 3c: Several mice on PBS showed increases in LVEF at baseline vs 4 weeks: why? Similar concern with LVESV: there was apparent improvement on PBS only.

Supplementary Fig 11: SERCA levels appear highly variable. Why?

Supplementary Fig 15 indicates super-rescue of F/F0 (Amplitude) and other Ca-kinetic parameters by PLN-ASO. What are the underlying mechanisms? Are there other subcellular alterations (besides SERCA inhibition) that are rescued in this model by PLN-ASO?

In the MLP-KO model: a) the rescue phenotype is assessed only at one week after the final injection. The Max dP/dt, Min dP/dt and tau do not appear to respond to dobutamine in the PLN-ASO model. In fact, they appear similar to the the MPL-KO. Why?

b) PLN-ASO: increased max dP/dT over control levels (over WT or +/-mice). Also all Ca-kinetics were super-rescued. Please address.

Rat model: PLN-ASO started at 6 weeks post-MI and 7 doses were administered up to 10 weeks. Then functional evaluation was done at 11 weeks. There is improvement of +dP/dt but not -dP/dt: Why? PLN-ASO is supposed to mainly affect relaxation in the heart. The dobutamine studies indicate that the relative increases in Fig 3d are not different between PLN-ASO and the other groups, indicating lack of rescue in this respect.

Some of the data are presented in both the manuscript Figures and the Supplemental Figures. Duplicates may be omitted.

REVIEWER COMMENTS

Reviewer #1 (Remarks to the Author):

The present manuscript is aimed to study the therapeutic use of antisense oligonucleotides (ASOs) targeting the expression of phospholamban (PLN) in cardiac diseases. The authors comprehensively addressed the effects of ASOs treatment in vitro as well as in vivo on three murine models of heart diseases, including two independent genetic models of dilated cardiomyopathy (DCM) and one of myocardial infarction (MI). Many ASOs were initially screened in vitro to evaluate the most efficient to downregulating the PLN expression and then, in vivo experiments were performed to select the ones with the least toxicity. Two PLN-ASOs were used in mice.

The overall work is well organized, compels a great amount of novel data, results very well presented and the methodology used is appropriate. The article is well referenced, and the conclusions are supported by the results.

Specific Concerns:

1. Although the studies in both models of DCM include intracellular calcium measurement either in neonatal or adult cardiomyocytes, it is important to study how the PLN-ASO treatment affects the calcium handling in the model of an acquired cardiomyopathy in this study, MI in rats. So, calcium measurements are recommended, especially in this model. Accordingly, the section of the study that addresses MI is evidently less exhaustively explored than the sections of DCM. For instance, SERCA2 expression (which is expected to be reduced in a HF model) is not showed. It would be very interesting to see, in this particular model, how the ratio SERCA2/PLN is affected by the treatment and its correlation with calcium decay rate, etc.

We agree with the reviewer that providing more insight into cardiomyocyte calcium handling in post MI rat cardiomyocytes will be informative. For this purpose, we have now performed a SERCA2 western blot and studied T-tubule organization.

We did not observe reduced SERCA2 protein levels in post-MI rats compared to sham operated rats in this study. This observation is consistent with the fact that the animals were in a moderate rather than advanced state of HF. Indeed, the average LVEF% was approximately 45% at 6 weeks post MI. Nevertheless, the 50mg/kg PLN ASO treatment significantly reduced PLN levels, resulting in an increased SERCA2 to PLN ratio (see below) which can account for the beneficial effects on cardiac contractility (dP/dt measurements) and cardiac dimensions seen in this model. We have added the SERCA2/PLN ratio data, as well as a representative WB image for SERCA2 protein to Fig.4b.

As we have now re-performed the western blot for PLN and SERCA2 to calculate the SERCA2/PLN ratio, we have replaced the western blot for total PLN included in the initial submission with the new blot. Therefore, also the complete western blot images of Fig. S20 are replaced by the new blots. The semi-quantification of SERCA2 protein levels, shown below, is added to Fig. S19.

Legend. Western blot results of LV protein lysates stained for PLN and SERCA2 protein and semi-quantified relative to PBS treated control samples, intensities are normalized to GAPDH (n=5 for sham, n=7 for PBS and PLN-ASO 50mg/kg and n=8 for control ASO and PLN-ASO 25mg/kg). One way analysis of variance is used for analyses, with PBS treated animals as the reference group in multiple comparison analyses. Asterix denotes significance level compared to PBS with: *<0.05, **<0.01, ***<0.001 and ****<0.0001. Single values are depicted, and error bars represent standard error of the mean (SEM).

It is well established that reduced SERCA expression and activity is a key mechanism for the progression to heart failure. However, sarcoplasmic Ca^{2+} release is also impaired in failing cells due to changes in the t-tubule organization. In healthy cardiomyocytes, t-tubules place L-type Ca^{2+} channels (LTCCs) in close proximity to ryanodine receptors, allowing efficient triggering of Ca^{2+} release. During heart failure, t-tubule loss results in the formation of “orphaned” ryanodine receptors, promoting delayed Ca^{2+} release and less powerful contraction (reviewed in PP Jones et al., *Fronts Physiol*, 2018).

Previous work has shown that reversal of heart failure induces parallel recovery of both, SERCA2 expression level and t-tubule organization (reviewed in Jones et al). Furthermore, in a previous study employing SERCA overexpression in heart failure, t-tubule integrity was observed to be improved (Lyon et al. *Circ Heart Fail*. 2012). These data suggest that there is an important link between declining SERCA2 expression levels, t-tubule structure, and systolic dysfunction. To investigate this hypothesis in our post-MI rats, we have now performed t-tubule analysis. In agreement with previous work, we observed that, compared to sham operated rats, placebo-treated post-MI animals exhibited reduced density of transversely-oriented t-tubules, which are the primary sites of Ca^{2+} -induced Ca^{2+} release in cardiomyocytes. This effect was significantly attenuated in animals treated with the high dose PLN-ASO, as t-tubules were re-established to levels similar to sham-operated controls. These data are now added to Fig. 4e, and discussed in the results section. These new data support that therapeutically increasing SERCA2 activity can reverse detrimental remodeling in the failing heart at both the whole-heart and subcellular level, improving cardiac performance.

Legend. Left ventricular sections from sham, PBS and PLN ASO 50mg/kg treated animals are stained for Caveolin-3 to visualize the transverse and longitudinal t-tubule organization. The density and proportions of transverse and longitudinally-oriented tubules were determined using a custom-made software in MatLab (see methods for details). Number of cells/animals analyzed: sham: n=83 (rats n=5), PBS n= 82 (rats n=5) , control ASO n=76 (rats n=4), PLN-ASO 25mg/kg n=35 (rats n=3), PLN-ASO 50mg/kg n=54 (rats n=5). A repeated measures model with one-sided paired contrasts is used to compare the mean differences between treatments and a control group (PBS). Dunnett's test is used to adjust p-values for multiple contrasts with the vehicle/reference group.

2. In line with the previous observations, the authors mention that the same screening process was carried out for selecting the most appropriate ASO for treating rats, however no data is shown. Moreover, they do not provide information about what particular PLN-ASO was administered to the rats. Please complete this missing information.

The reviewer is correct that these data should be represented in the manuscript. We now added this as Fig. S18, see panels below. Similar to the mouse PLN-ASO screen, we started off with 312 rat-specific PLN-ASOs screened initially at a single concentration of 3 µM in rat L2-RYC cells following electroporation and 24 hours of incubation. L2-RYC cells are rat yolk sac carcinoma cells that express PLN. These were deemed as reliable as rat primary cardiomyocytes for ASO screens, and were chosen given the ease and reproducibility of working with immortalized cells vs. cardiomyocytes. Below in panel B are the top 36 PLN-ASOs which demonstrated ~70% reduction of *Pln* mRNA and were taken into an in vitro dose-response comparison. PLN knockdown compared to untreated controls (UTC) was assessed using PLN-ASO doses ranging from 333nM to 9µM, also in rat L2-RYC cells following electroporation and 24 hours of incubation. ASOs with ≤ ~1 µM IC₅₀ (21 total as marked) were chosen for large scale synthesis to support in vivo evaluation.

b

Legend. (a) Schematic illustration of experimental steps leading to identification of the top rat PLN ASO for in vivo studies. (b) Results of secondary in vitro screen illustrating *Pln* mRNA reductions in a dose response comparison. ASO candidates are selected (21) if dose-response is demonstrated and *Pln* mRNA is reduced more than 50% in rat L2-RYC 24h after treatment. (c) Selection of top ASO from in vivo screen in healthy Lewis rats based on a consideration of heart *Pln* mRNA reductions and plasma ALT and AST after 3 s.c. injections of 50mg/kg. One-way analyses of variance are used. Asterisk denotes significance level compared to PBS with: * <0.05 , ** <0.01 , *** <0.001 and **** <0.0001 .

An in vivo screen was performed in normal Sprague Dawley rats to validate target engagement and screen for liver toxicity using plasma ALT and AST levels following SC injections of 50 mg/kg/week given over 3 weeks. Using these data (see below), we selected ASO#136 as our candidate ASO for future rat studies.

C

3. Why the ASO control was not used (or showed if experiments were done) in the mouse models of DCM?

An ASO control group was utilized in two of the three models (*Cspr3/Mlp*^{-/-} and rat MI) and we include now a detailed analyses of control ASO effects in these models. In the two studies where a control ASO (Ctrl-ASO) was included, no significant differences were observed between control ASO (Ctrl-ASO) and vehicle control treatment (PBS). For the rat MI study, this can be observed in Fig. 4 and below, where cardiac PLN protein levels, echocardiography parameters and invasive hemodynamics show no differences between control ASO (Ctrl-ASO) and vehicle control (PBS).

Legend. Western blot results of LV protein lysates stained for PLN protein and semi-quantified relative to PBS treated control samples, intensities are normalized to GAPDH (n=5 for sham, n=7 for PBS and PLN-ASO 50mg/kg and n=8 for control ASO and PLN-ASO 25mg/kg). Representative image of 1 out of 4 membrane stains (Fig. S20).

Legend. Individual MRI assessment and quantification of the change in left ventricular ejection fraction (LVEF), left ventricular end systolic volume (LVESV) and left ventricular end diastolic volume (LVEDV) between treatment initiation (6 weeks post MI) and 5 weeks after start of treatment showing (n=5 for sham, n=7 for PBS, n=8 for control ASO, n=8 for PLN-ASO 25mg/kg and n=6 for PLN-ASO 50mg/kg).

Legend. Hemodynamic assessment of the maximum/minimum first derivative of LV pressure (Max dP/dt and Min dP/dt) performed at study end, 5 weeks after treatment start, at baseline and upon increasing dobutamine doses of 0, 1, 2 and 6 $\mu\text{g}/\text{kg}/\text{min}$ (n=5 for sham, n=7 for PBS and PLN-ASO 50mg/kg and n=8 for control ASO and PLN-ASO 25mg/kg).

We already had performed a *Cspr3/Mip*^{-/-} PLN ASO study (study II), with a similar study design as the study presented in Figure 3 (see below). Cardiac functional measurements were actually included in the supplementary table 1 at the initial submission. Either control ASO or ASO #26_C were subcutaneously injected at baseline (100mg/kg), and week 1 and 2 (50mg/kg) in adult *Cspr3/Mip*^{-/-} male and female mice. At week 6 post treatment start, echocardiographic analyses were performed.

Legend. Experimental design of the PLN-ASO *Cspr3/Mip*^{-/-} intervention study. Study was performed with PLN-ASO #26_C (100mg/kg on day0, 50mg/kg on day7 and 14) and control ASO. Echocardiography was performed at baseline (i.e. before treatment initiation), and at end of study (i.e. after 42 days of treatment)

The results of the study can be observed below and are now added in the results section and as Fig. S17 to the manuscript. These data show a significant improvement in LVEF, LVESV and a positive trend towards improvement in LVEDV between PLN-ASO treated and control ASO treated *Cspr3/Mip*^{-/-} mice. Also see Supplementary Table 1 for the individual echocardiography measurements of all *Cspr3/Mip*^{-/-} PLN ASO studies.

Legend. Individual echocardiography assessment and quantification of left ventricular ejection fraction [LVEF], LV end-systolic volume (LVESV), and LV end-diastolic volume (LVEDV) 28 days after treatment relative to baseline measurements. (PLN-ASO #26_C treated n=10, Control-ASO n=6). Students T test is used for analyses. Asterix denotes significance level compared to control ASO with: * <0.05 , ** <0.01 , *** <0.001 and **** <0.0001 . Single values are depicted, and error bars represent standard error of the mean (SEM).

To prevent an unfair comparison (that potential beneficial effects of the PLN-ASO might be exaggerated because of any detrimental effects of the control ASO), we chose to prioritize a vehicle control treated group over a control ASO treated group in future studies. This agrees with several ASO studies recently published (for example: Foinquinos et al. Nature Communications 2020, Ran et al. EMBO Molecular Medicine 2020) and with a design that would be used in coming clinical trials (Täubel et al. Eur Heart J. 2020). Therefore, the PLN R14del study utilized a vehicle control group. In this study, the PLN-ASO improved survival over 3-fold. Considering the effects of control ASO's in our studies and studies performed by other groups, we consider it highly unlikely that a control ASO would result in such a substantial survival benefit. We trust that our data with a control ASO in two of the three models, *Cspr3/Mip*^{-/-} and rat MI, are sufficient to convince the reader of the cardiac specific effects of the PLN-ASO to reduce PLN expression and not due to a class effect of the ASO to increase SERCA2a activity or improve calcium homeostasis.

Minor comments

1. Figure 2h, the ECG trace for the PLN R14 $\Delta\Delta$ +PBS is not representative of the averaged values for waves R and S, and exaggerates the differences with the other groups.

We have replaced the ECG tracing by a more conservative tracing, see Fig. 2 and below. No available ECG tracing matches 100% with the averages: the now presented tracing shows slightly higher amplitudes for the PBS treated group.

2. Figure 3e, all 4 subpanels are repeated in the Suppl. Figure 13. Please, be sure to use the supplemental figures to show only supporting data and not duplicate them.

Duplicates have been removed from the supplementary figures as suggested.

3. Figure 4 is full of panels which could be better compiled in a table or a lesser number of panels, and move most of them to the supplemental figures.

We agree with the reviewer's comment. Therefore, we have removed the individual line tracings over time (Fig. 4c), and only show the "change from baseline" graphs instead. The same was applied to Fig. 3c. Please find the new panel 3c and 4c below.

Panel 3C.

Panel 4C.

4. Supplemental Figure 15a, the color code of the calcium transient traces seems to be switched with respect to the rest of the figure/s. It would be easier for the reader if the colors of the traces match with the ones in the bars.

This is now corrected in the figure, please also see below.

5. In Supplemental Figure 15e and f the authors show APD50 and APD75, respectively. APD is not defined, but instinctively one could guess Action Potential Duration. However, the authors have not described any electrophysiology experiment measuring membrane potential of action potential. Moreover, the legend for the Suppl. Fig. 15 says: ADP50 and ADP75. Could you please clarify what parameter is depicted here?

This is well noted by the reviewer. Indeed, we did not perform electrophysiology experiments, the depicted data concern calcium imaging measurements and should therefore be labelled as “time to base 50% or 75%”, referring to calcium decay which is compared to the WT group. The graphs are updated accordingly and depicted in Fig. S16 and below.

Reviewer #2 (Remarks to the Author):

This manuscript describes the activity of antisense oligonucleotides in cardiac tissue. Relatively little good work has been done investigating the activity of ASOs in the heart, making this a novel topic.

Unfortunately, there are two big drawbacks that cannot be overlooked.

1) There is no detailed description of the assays using ASOs nor any information about their composition. The exact molecular structure of the compounds needs to be disclosed.

The reviewer is right and we provide now the exact molecular structure as requested. ASOs used in this report were 16mer (S)-constrained ethyl (cEt) modified 3-10-3 gapmers containing three cEt-modified ribonucleotides in each end and 2'-deoxynucleotides in the middle portion of the molecule with a phosphorothioate backbone as reviewed in Crooke ST, Witztum JL, Bennett CF, Baker BF. RNA-Targeted Therapeutics. Cell Metab. 2018 Apr 3;27(4):714-739. This is now mentioned in the introduction and more extensively in the methods section.

2) PBS is not an adequate control. For studies like these control oligonucleotides of similar sequence but with disrupted base-pairing must be used.

An ASO control group was utilized in two of the three models (*Cspr3/Mip*^{-/-} and rat MI) and we include now a detailed analyses of control ASO effects in these models. In the two studies where a control ASO (Ctrl-ASO) was included, no significant differences were observed between control ASO (Ctrl-ASO) and vehicle control treatment (PBS). For the rat MI study, this can be observed in Fig. 4 and below, where cardiac PLN protein levels, echocardiography parameters and invasive hemodynamics show no differences between control ASO (Ctrl-ASO) and vehicle control (PBS).

Legend. Western blot results of LV protein lysates stained for PLN protein and semi-quantified relative to PBS treated control samples, intensities are normalized to GAPDH (n=5 for sham, n=7 for PBS and PLN-ASO 50mg/kg and n=8 for control ASO and PLN-ASO 25mg/kg). Representative image of 1 out of 4 membrane stains (Fig. S20).

Legend. Individual MRI assessment and quantification of the change in left ventricular ejection fraction (LVEF), left ventricular end systolic volume (LVESV) and left ventricular end diastolic volume (LVEDV) between treatment initiation (6 weeks post MI) and 5 weeks after start of treatment showing (n=5 for sham, n=7 for PBS, n=8 for control ASO, n=8 for PLN-ASO 25mg/kg and n=6 for PLN-ASO 50mg/kg).

Legend. Hemodynamic assessment of the maximum/minimum first derivative of LV pressure (Max dP/dt and Min dP/dt) performed at study end, 5 weeks after treatment start, at baseline and upon increasing dobutamine doses of 0, 1, 2 and 6 $\mu\text{g}/\text{kg}/\text{min}$ (n=5 for sham, n=7 for PBS and PLN-ASO 50mg/kg and n=8 for control ASO and PLN-ASO 25mg/kg).

We already had performed a *Cspr3/Mip*^{-/-} PLN ASO study (study II), with a similar study design as the study presented in Figure 3 (see below). Cardiac functional measurements were actually included in the supplementary table 1 at the initial submission. Either control ASO or ASO #26_C were subcutaneously injected at baseline (100mg/kg), and week 1 and 2 (50mg/kg) in adult *Cspr3/Mip*^{-/-} male and female mice. At week 6 post treatment start, echocardiographic analyses were performed.

Legend. Experimental design of the PLN-ASO *Cspr3/Mip*^{-/-} intervention study. Study was performed with PLN-ASO #26_C (100mg/kg on day0, 50mg/kg on day7 and 14) and control ASO. Echocardiography was performed at baseline (i.e. before treatment initiation), and at end of study (i.e. after 42 days of treatment)

The results of the study can be observed below and are now added in the results section and as Fig. S17 to the manuscript. These data show a significant improvement in LVEF, LVESV and a positive trend towards improvement in LVEDV between PLN-ASO treated and control ASO treated *Cspr3/Mip*^{-/-} mice. Also see Supplementary Table 1 for the individual echocardiography measurements of all *Cspr3/Mip*^{-/-} PLN ASO studies.

Legend. Individual echocardiography assessment and quantification of left ventricular ejection fraction [LVEF], LV end-systolic volume (LVESV), and LV end-diastolic volume (LVEDV) 28 days after treatment relative to baseline measurements. (PLN-ASO #26_C treated n=10, Control-ASO n=6). Students T test is used for analyses. Asterix denotes significance level compared to control ASO with: *<0.05, **<0.01, ***<0.001 and ****<0.0001. Single values are depicted, and error bars represent standard error of the mean (SEM).

To prevent an unfair comparison (that potential beneficial effects of the PLN-ASO might be exaggerated because of any detrimental effects of the control ASO), we chose to prioritize a vehicle control treated group over a control ASO treated group in future studies. This agrees with several ASO studies recently published (for example: Foinquinos et al. Nature Communications 2020, Ran et al. EMBO Molecular Medicine 2020) and with a design that would be used in coming clinical trials (Täubel et al. Eur Heart J. 2020). Therefore, the PLN R14del study utilized a vehicle control group. In this study, the PLN-ASO improved survival over 3-fold. Considering the effects of control ASO's in our studies and studies performed by other groups, we consider it highly unlikely that a control ASO would result in such a substantial survival benefit. We trust that our data with a control ASO in two of the three models, *Cspr3/Mip*^{-/-} and rat MI, are sufficient to convince the reader of the cardiac specific effects of the PLN-ASO to reduce PLN expression and not due to a class effect of the ASO to increase SERCA2a activity or improve calcium homeostasis.

Point 1) can be easily corrected. Unfortunately, while 2) could have been easily addressed when these experiments were done, it would be necessary to now redo the experiments with proper controls. Sadly, because activity in the heart would be novel, I do not believe that the lack of accepted rigor can be excused in this case.

I appreciate the heart work that went into this study, but the flawed experimental design reduces my enthusiasm for the work.

Reviewer #3 (Remarks to the Author):

This study describes the rescue of three heart failure models by antisense phospholamban oligonucleotides (PLN-ASO). Two of the models were genetic mouse models of HF and the third one was an MI rat model of HF. Previous studies have supported the benefits of targeting the impaired Ca-cycling in heart failure through increasing either the SERCA2a levels or decreasing the inhibitory levels/activity of PLN. Initially 312 ASOs were evaluated by the authors, 21 were selected for further evaluation in healthy mice (3 subcutaneous administrations) and finally two ASOs were selected for evaluation in the HF models. Comprehensive studies were performed in each model using integrative approaches to evaluate the benefits of PLN-ASO. The first model is the PLN-R14del mouse and PLN-ASO prevented PLN aggregates, cardiac dysfunction and extended the life span of the animals. The second model is the *Cspr3*/MLP-KO mouse and PLN-ASO reversed the HF phenotype. The third model is a rat MI model where PLN-ASO improved contractility.

Concerns

Overall, the benefits associated with PLN-ASO are only observed for a very short time after the final dose administration, making the therapeutic potential of this approach rather limiting. Several ASO doses are administered in each of the models and studies are mainly performed within a week after the final dose administration. Longer term effects are not assessed except for the survival study in the PLN-R14del model (but PLN-ASO had to be administered throughout the life span of the model).

We thank the reviewer for the careful consideration of our work. Indeed, ASOs are a modality that need repeated administrations. This is in line with the mode of action, which is to prevent messenger RNA translation, and not gene correction, as well as the duration of action which is on the order of weeks. All studies performed with ASOs require repeated administrations, including ASOs used in the clinic. These include, amongst others, Inotersen (Tegsedi) used for hereditary transthyretin amyloidosis which requires weekly subcutaneous injections and Nusinersen (Spinraza) used for spinal muscular atrophy requiring repeated intrathecal administrations. We see no reason to believe that this would be different in the context of cardiovascular disease, especially hereditary cardiomyopathies. Since the factor inflicting the damage (MLP or PLN mutation in this specific case) is unaltered, a chronic downregulation of PLN mRNA would be required. Therefore, ASOs have been developed with improved nuclease resistance and longer half-lives. To emphasize this feature of ASO therapy, which is also an advantage as dosing can be adjusted over time, we have added "Importantly, like other HF standard of care medications, ASOs require repeated administrations, doses can be titrated and thus dose-responsive pharmacology achieved, unlike AAV mediated gene therapy." to the discussion. Additionally, we added data (see below) in Fig. S13 detailing the onset and duration of PLN mRNA and protein down-regulation after PLN-ASO treatment in *Cspr3*/MLP-/- mice. These data show a lag between *Pln* mRNA vs. protein reductions, as would be expected given the shorter half-life of mRNA vs. protein and the mechanism of ASO therapy. Additionally, these data reveal a sustained PLN ASO treatment effect, as reduced *Pln* mRNA and protein levels are maintained for weeks after treatment cessation with protein reductions remaining suppressed even longer. Initial repeated dosing is used to faster achieve a steady-state of PLN down-regulation. Such data will be used to help inform the optimal therapeutic dosing of the human drug.

Legend. PLN protein and mRNA levels, depicted as % of PBS. For this study, 8 to 12-week-old female *Cspr3*/MLP-/- mice were used. N = 2/time point/dose level. 30 or 100 mg/kg PLN ASO was subcutaneously injected on days 0, 2, 4, 7, 14 and 21. Cohorts euthanized on Days 4, 7, 14, 21, 35, 63, 91 and 119. Mice euthanized on Days 4, 7, 14 and 21 received a total of 2, 3, 4, and 5 ASO doses, respectively. All other cohorts (i.e. mice euthanized on Days 35, 63, 91 and 119) received 6 doses.

The PLN-R14del model used in this study is homozygous for the R14del-PLN mutation but the patients are heterozygotes. This model shows a severe phenotype by 8-9 weeks of age. It would be more appropriate to carry out this study with PLN-ASO in the heterozygous model with the same genotype as the R14del-patients.

We agree with the reviewer's comment. However, heterozygous mice only present the first signs of cardiomyopathy at the age of 18 months: impaired cardiac contractile function, increased myocardial fibrosis and the presence of PLN protein aggregates, thus such a study would span over 2 years. Therefore, the homozygous PLN-R14del mouse model was utilized in this study as this is considered an accelerated model of the phenotype observed in heterozygous PLN-R14del mice (Eijgenraam et al. Sci Rep. 2020). We believe that further confirmation of therapy efficacy and safety would not be obtained with an additional study in heterozygous mice but require large animal models and ultimately human phase 1 studies.

Fig 1e: The PLN mRNA remains low only up to Day 14 following the 3 doses. It increases after day 14 post the third injection/dose. Indeed, the PLN protein (Suppl Fig 1a) shows increases up to 50% of PBS levels by day 42. Thus, the decreases in PLN mRNA or protein are temporary and maintenance of PLN-ASO administration would be a requirement in clinical studies. Please address.

We completely agree, repeated administrations are an absolute requirement as first mRNA levels, and later also protein levels slowly increase back to 100% (see also answer to your first comment). ASOs have a half-life, determined by their chemistry. This can also be observed in the pharmacokinetics/dynamics study presented in the first question and as Fig. S13. Further, this is now also emphasized at the end of the discussion:

"Importantly, like other HF standard of care medications, ASOs require repeated administrations, can be titrated and thus controlled, unlike AAV mediated gene therapy."

Fig 1f: neonatal myocytes are used but these studies should be performed in adult or mature cardiomyocytes. Is SR Ca-load altered by PLN-ASO? Why isn't there an increase in FS when the Ca-peak was increased? Are there any alterations in myofilament activity?

The cardiomyocyte studies shown in Fig 1. are indeed performed in neonatal cardiomyocytes as they had the main objective of screening for the optimal ASO. Since very large cell numbers were required for this purpose, we chose to use mouse neonatal cardiomyocytes. The reviewer is completely right that mechanistic studies are best performed in mature cardiomyocytes or adult heart. Therefore, we analyzed the effect of the PLN-ASO on calcium cycling in adult cardiomyocytes isolated from *Cspr3/Mlp*^{-/-} mice, please see the figure on the next page and Fig. S16). These data showed an increased calcium amplitude with increased calcium upstroke and decay velocities. As for Fig. 1f., we hypothesize that the lack of change in fractional shortening in the neonatal cardiomyocytes is indeed due to the immaturity of the cardiomyocytes, which coincides with immature expression patterns of calcium handling proteins, such as SERCA2a. The contribution of SERCA2a to calcium cycling is therefore likely modest and correspondingly, the change in calcium peak caused by the PLN-ASO is also relatively modest (F_{max}/F₀ ratio 1.04 to 1.08). The observed increase (1.04 to 1.08) might very well be insufficient to result in an increased fractional shortening. In comparison, the PLN-ASO in adult cardiomyocytes of the *Cspr3/Mlp*^{-/-} model resulted in an increase of the calcium peak of 1.52 to 4.58 (F_{max}/F₀ ratio, P<0.05).

* Comparison to *Cspr3/Mlp^{+/+}*; * Comparison to *Cspr3/Mlp^{-/-}* without PLN-ASO

Legend. Imaging of whole cell intracellular calcium flux and analysis was performed on adult cardiomyocytes isolated 4 weeks after treatment from hearts of *Cspr3/Mlp^{+/+}*, *Cspr3/Mlp^{-/-}* untreated and PLN-ASO treated mice (as described in Fig. 3a). Cardiomyocytes were incubated with Fluo-4 before imaging and paced at 1Hz. Calcium transients are calculated as $\Delta F/F_0$ (F_0 is background fluorescence, F is fluorescence intensity). (a) Representative traces of intracellular calcium transients comparing 3 conditions are shown. For each cell, (b) Amplitude (peak intensity), (c) Upstroke Velocity, (d) Decay Velocity, (e) Time to base 50%, and (f) Time to base 75% were calculated. PLN-ASO treatment significantly enhanced amplitude (surrogate for force), upstroke (surrogate for contraction/increased RyR2 calcium release kinetics) and decay velocity (surrogate for relaxation / SERCA2a activity) compared to untreated *Cspr3/Mlp^{-/-}* cardiomyocytes which is in line with PLN-ASO effects observed in vivo. Total cell number measured from 3 independent experiments for *Cspr3/Mlp^{-/-}* and 2 for *Cspr3/Mlp^{+/+}*: *Cspr3/Mlp^{+/+}* n=286; *Cspr3/Mlp^{-/-}* n=146, *Cspr3/Mlp^{-/-}* PLN ASO n=262. Statistics: Pairwise comparisons, two-way analysis of variance is used for analyses and p-values were corrected using Tukey's procedure. Asterisk denotes significance level (comparisons indicated in Figure): * <0.05 , ** <0.01 , and *** <0.001 , error bars represent standard error of the mean (SEM).

In the R14del-model: there are 4 doses at weeks 3-6 and then assessment is done a week later (at 7 weeks of age). What are the specific Ca-defects that PLN-ASO targets and restores in this model?

This is a very interesting question, which is highly debated in the literature. There are 2 prevailing hypotheses by which the PLN-R14del mutant protein could cause cardiomyocyte dysfunction. The first is by an altered interaction with SERCA2a, the second by toxic PLN protein aggregation resulting in ER stress, cardiomyocyte dysfunction, death and fibrofatty replacement. Several papers studied the effects of the PLN-R14del protein on its interaction with SERCA2a and their results depend on the model and methods utilized. In enzyme assays, the mutant impaired PLN phosphorylation, thereby theoretically increasing SERCA2a inhibition (Kim et al. PNAS 2015 and Hughes et al. Plos One 2014). In stark contrast, in lipid vesicles and HEK-293 cells, PLN-R14del resulted in less inhibition of SERCA2a (Vostrikov et al. Biochim et Biophys Acta 2015 and Haghghi et al. PNAS 2006). Finally, in mice with PLN-R14del cDNA introduced and crossed with PLN-knockout mice, no differences were observed between PLN-R14del and WT PLN on SERCA2a activity assays. Our RNAseq results did not reveal calcium related pathways to be improved by the PLN-ASO (Fig S8). Instead, protein processing pathways and the unfolded protein response were amongst the top affected pathways. Together with immunofluorescence stainings (Fig 2d.) and western blots (Fig S5.) showing reduced PLN protein aggregates, we hypothesize that the mechanism by which the PLN-ASO positively affects cardiac function and survival is at least in part through reducing toxic PLN protein aggregation. This is mentioned in the results section as follows:

“These findings provide further support to the idea that a reduction of toxic PLN protein aggregates by PLN-ASO might contribute to the significant benefits observed after treatment.”

The R14del-PLN model shows 35% of mRNA levels compared to WT and 40% of PLN protein levels compared to WT (PBS data). Thus, if PLN is decreased, why does one need to further decrease it by PLN-ASO? If the low levels of PLN protein are due to its presence in aggregates, then one would not expect to observe similar decreases in PLN mRNA levels in this model (PBS data: Fig 2b).

When designing this study, we hypothesized that we could improve cardiac function and survival by reducing the PLN aggregates using the PLN-ASO. Clearly, the results of the study indicate that the physiological downregulation of PLN in the homozygous PLN-R14del model is insufficient, as still aggregates are formed and a cardiomyopathy develops, which can be reversed by further downregulation of PLN mRNA. The answer to why PLN mRNA and protein levels are already decreased at baseline likely relates to the pathophysiology of the PLN-R14del protein. In the Western Blot of Fig 2c. (also see below), we combined non-aggregated and aggregated protein samples in ratio (dissolvable in Urea, see also next response). Therefore, this quantifies the total PLN protein level and the reduction observed in PLN protein levels (Fig. 2c) is not caused by a transformation into aggregates, and there is a true reduction in PLN protein levels in these mice. Consequently, another feedback mechanism is probably the causative factor for the lower mRNA expression levels and lower protein levels of PLN (Fig. 2b and c). Theoretically, this could be an increased inhibition of SERCA2a by the mutant PLN.

Supplemental Fig 3 shows a full blot. Which bands are used to quantitate PLN in Fig 2C? How is the PLN level quantitated in Suppl Fig 5? It is expressed as fold change (To PBS) but this is not clear. What does the PLN data in “Dissolvable in RIPA” vs “Dissolvable in UREA” mean? Which of these data in Suppl Fig 5 correspond to the data in Fig 2c?

Protein lysates were obtained using RIPA, and the remaining protein pellet after RIPA was dissolved in urea. Urea is a well-known strong denaturant for proteins and therefore widely used for protein unfolding. This fraction of protein that is only dissolvable in urea is considered the fraction representing the aggregated proteins. For Fig. 2c and Fig. S2, these 2 fractions (RIPA and Urea) are added together in ratio. Fig. 2c is based on the full blot of Fig. S2. Both the 25kda and 5kda bands (pentamers and monomers) are quantified and added together to get to the total PLN protein levels depicted in Fig. 2c. Fig. S5 is based on the full blots of Fig. S3 and S4, and does not correspond to the data of Fig.2c. In this distinct blot, the RIPA and Urea fractions were analyzed separately to be able to quantify the effect on the presumed aggregated fraction. For the results, monomers, dimers and pentamers were quantified and represented in Fig. S5.

Fig 2d: Please use arrows to point to the aggregates.

White arrows have been added to Fig 2d. and the legends.

The mouse survival study required a lower maintenance dose every 4 weeks. In this follow-up, the PLN mRNA levels increased at T15-23 compared to PLN-ASO (Suppl Fig 10a) but the PLN protein levels remained low through T23 (Supl Fig 10b). How do the authors explain this apparent discrepancy between mRNA and protein levels?

The Pln mRNA and protein analysis 4 weeks after PLN ASO treatment initiation (after 4 doses), shows an effective downregulation of Pln mRNA and protein levels in the heart. The fact that increased Pln mRNA levels, but not protein levels, are observed at T15-23 is due to the fact that there is generally a delayed response seen on the protein level following changes on the mRNA level (this is visualized in Fig. S13 and mentioned in the response to your first question). This observation further suggests that a higher or more frequent maintenance dose will be required to keep mRNA levels low (and in consequence also protein levels). Likely, this might have an additional beneficial effect on survival and/or cardiac function. However, interpretations in the PLN R14del study have to be made with extreme caution, as we do not have any untreated control as comparison at this stage as all untreated mice have died at 9 weeks of age.

Also, in Supplemental Fig 10:

- a) **Although PLN-ASO was administered every 4 weeks, fibrosis increased starting at T19. At T23, fibrosis was similar to PBS: why?**
- b) **LVEF and LVESV showed deterioration starting at T19 and at T23, they were similar to PBS. Please discuss.**
- c) **Similar concerns exist with the ECG: R, S and R+S approached PBS levels by T19 and T23. Please address.**

All these observations are explained by the fact that we administrated very low doses of the PLN-ASO on purpose. Based on pharmacokinetics (Fig. S13), we expected that a once weekly dose of 50mg/kg was required to continue the achieved level of PLN downregulation. By using a once monthly dose of 50mg/kg, we expected to observe a slow return towards disease phenotype, which could provide us clues towards uncovering disease pathophysiology and the mechanism of action of the PLN-ASO. Indeed, at T19, mainly fibrosis and LVEDV show deterioration, identifying these as early mechanisms of disease comparable to what is observed in human PLN R14del patients (Te Rijdt et al. Eur Heart J Cardiovasc Imaging, 2019). This is in contrast to other types of cardiomyopathy, where fibrosis often is a late stage feature. In addition, this strategy allowed us to study the RNAseq patterns in more detail (Fig. S8.) This also highlights, as discussed in the first reviewer question, that ASO treatment indeed requires repeated administration with an adequate dose.

Figure 3c: Several mice on PBS showed increases in LVEF at baseline vs 4 weeks: why? Similar concern with LVESV: there was apparent improvement on PBS only.

A prior report evaluated age-dependent changes in LVEF in MLP KO mice (Costandi PN, Frank LR, McCulloch AD, Omens JH. Role of diastolic properties in the transition to failure in a mouse model of the cardiac dilatation. Am J Physiol Heart Circ Physiol. 2006 Dec;291(6):H2971-9). Shown below are Figure 1 data from that article that show a compensatory improvement in LVEF during a phase of rapid anatomical remodeling. Although the MLP KO mice we used were bred into a different background, it is plausible that similar phenomena could be occurring. Nonetheless, our data demonstrate PLN inhibition to have a robust effect to improve LVEF and chamber dilation despite any such compensatory changes.

Fig. 1.

MRI-derived time courses of remodeling for muscle LIM-only protein-deficient (MLP^{-/-}) vs. 129/Sv hearts. One-hundred forty-two MRI experiments from weekly imaging protocol are summarized at 18-binned time points. Plots are presented as means \pm SE of all mice imaged at nearest 3 time points, i.e., 2–4 wk, 5–7 wk, etc., for each genotype. Note the presence of hypertrophy up to 15 wk, as indicated by a comparable dilatation index among strains (A) but a larger end-diastolic volume (EDV) (B) in MLP^{-/-} hearts, and abrupt dilatation phase (indicated by bracket in A and B) that occurs temporally concurrent with increase in stroke volume (C) and ejection fraction (D). LV, left ventricular; Epi, epicardial. † $P < 0.05$, MLP^{-/-} vs. 129/Sv; ‡ $P < 0.0001$ 15- vs 31-wk MLP^{-/-}.

Supplementary Fig 11: SERCA levels appear highly variable. Why?

As shown in Fig. 3b, there is no significant difference of SERCA2 levels between PLN ASO and PBS treated animals after the normalization to GAPDH. The variation is also visible in the GAPDH bands. Likely, none equivalent protein amounts were loaded on the gel, even though the protein concentration was measured beforehand.

Supplementary Fig 15 indicates super-rescue of F/F0 (Amplitude) and other Ca-kinetic parameters by PLN-ASO. What are the underlying mechanisms? Are there other subcellular alterations (besides SERCA inhibition) that are rescued in this model by PLN-ASO?

The primary mechanism is an almost complete removal of SERCA2a inhibition by PLN. In a baseline state, SERCA2a is partly inhibited by PLN in mice. Downregulation of PLN to $\pm 20\%$ of normal protein levels is likely to be causative of a super-rescue of calcium cycling, as is observed in this study. Other subcellular alterations as a primary effect of the ASO are less likely as ASOs are very specific to their target. Therefore, we expect other beneficial effects to be secondary to the increased calcium cycling.

In the MLP-KO model: a) the rescue phenotype is assessed only at one week after the final injection. The Max dP/dt, Min dP/dt and tau do not appear to respond to dobutamine in the PLN-ASO model. In fact, they appear similar to the MPL-KO. Why?

Since the hemodynamic assessment was a terminal procedure, interim measurements could not be obtained. Prior reports (See selected Figures below) show homozygous or heterozygous PLN deletion to sharply attenuate

changes in LV contractility and relaxation with β -adrenergic agonism. Given PLN reductions of up to 90% in our studies, our data are consistent with a primary role of PLN to repress basal contractility and PLN de-repression to be a key mechanism in the contractile response to β -adrenergic agonists.

FIG 4. Graphs showing lack of isoproterenol-stimulatory effects in phospholamban-deficient mouse hearts. Hearts from phospholamban-deficient (▲) and wild-type (■) littermates were perfused with increasing concentrations of isoproterenol. TPP indicates time to peak pressure; RT_{1/2}, half-relaxation time. The effects of isoproterenol on contraction (a and b), relaxation (c and d), and heart rate (e) are shown. Values are mean \pm SEM of six different hearts.

From Luo W, Grupp IL, Harrer J, Ponniah S, Grupp G, Duffy JJ, Doetschman T, Kranias EG. Targeted ablation of the phospholamban gene is associated with markedly enhanced myocardial contractility and loss of beta-agonist stimulation. *Circ Res.* 1994 Sep;75(3):401-9.

Fig. 3. Iso dose-response relationships for dP/dt_{max} , dP/dt at 40 mmHg developed pressure (dP/dt_{40}), and dP/dt_{max} divided by developed pressure at dP/dt_{max} ($dP/dt_{max}/DP$) in wild-type ($n = 7$), PLB heterozygous ($n = 8$), and PLB homozygous ($n = 5$) mice. Values for each animal were determined from at least 50 consecutive beats during final 30 s of each 3-min dose. Values are means \pm SE. *Significant effect compared with corresponding value in other 2 groups; † significant Iso dose effect; ‡ significant group-by-dose interaction.

From Lorenz JN, Kranias EG. Regulatory effects of phospholamban on cardiac function in intact mice. *Am J Physiol.* 1997 Dec;273(6):H2826-31.

b) PLN-ASO: increased max dP/dT over control levels (over WT or +/-mice). Also all Ca-kinetics were super-rescued. Please address.

In normal physiological conditions, SERCA2a is inhibited by PLN, resulting in inhibited Ca^{2+} -kinetics. Relieving this inhibition, by downregulating PLN expression, results in increased Ca^{2+} -kinetics. This is not only observed in disease models (Fig. S16), but in healthy cardiomyocytes as well (Fig. 1f). The increased Ca^{2+} -kinetics are well known to correlate to increased contractility (dP/dT), as calcium is responsible for the excitation-contraction (EC) coupling. Improved diastolic Ca^{2+} uptake results in higher sarcoplasmic reticulum Ca^{2+} stores and improved RyR activity (Fearnley et al. *Cold Spring Harb Perspect Biol* 2011). Together this effect results in increased force generation, as illustrated in this disease model.

Rat model: PLN-ASO started at 6 weeks post-MI and 7 doses were administered up to 10 weeks. Then functional evaluation was done at 11 weeks. There is improvement of +dP/dt but not -dP/dt: Why? PLN-ASO is supposed to mainly affect relaxation in the heart. The dobutamine studies indicate that the relative increases in Fig 4d are not different between PLN-ASO and the other groups, indicating lack of rescue in this respect.

Improvements in diastolic calcium uptake directly result in increased diastolic calcium stores and increased systolic calcium fluxes. Systolic calcium fluxes strongly correlate to contractility and max dP/dt , explaining the increased contractility caused by the PLN-ASO. In contrast, min dP/dt , or diastolic relaxation, is known to be impacted by a multitude of factors, besides diastolic calcium uptake, also including cardiomyocyte morphology (increased diameter in diastolic dysfunction) and tissue composition (interstitial fibrosis in diastolic dysfunction) (Borbely et al. *Circulation* 2005, van Heerebeek et al. *Circulation* 2006, Zile et al. *Circulation* 2015). We might speculate that this

could explain the improvement in calcium flux improved systolic dysfunction, but not diastolic dysfunction as the ischemia might have impaired cardiomyocyte and cardiac tissue integrity to an extent that could not be reversed with improved calcium homeostasis. Furthermore, new data added during revision have indicated that PLN-ASO treatment reverses t-tubule disruption during heart failure. This effect likely synergizes with augmentation of SR Ca²⁺ load to normalize to systolic dysfunction, while the links between t-tubule structure and diastolic function are generally not marked (Louch et al. Physiology, 2012).

Some of the data are presented in both the manuscript Figures and the Supplemental Figures. Duplicates may be omitted.

The following panels have been removed from the supplementary figures:

- Fig. S6 panel A
- Fig. S11, first 3 panels
- Fig. S14, min and max dP/dT panels
- Fig. S21, min and max dP/dT panels

REVIEWER COMMENTS

Reviewer #1 (Remarks to the Author):

The authors have properly addressed all relevant concerns. They added new data in agreement with the reviewers' suggestions I consider that the revised version of the manuscript is improved.

I have no further concerns.

Reviewer #2 (Remarks to the Author):

I appreciate the authors response.

There have been hundreds of similar papers published claiming ASO-mediated effects in extra-hepatic tissues. The work in the CNS has been convincing, other tissues, not so much. The controls used here remain insufficient to distinguish this work from many other papers. Action in the heart needs to be treated as a remarkable result that needs more than routine support. I cannot support this manuscript but I will understand if editors feel otherwise. It is certainly true that similar papers are routinely published.

There would need to be at least one new experiment using multiple control and multiple target-directed oligonucleotides in vivo to set this submission apart. I'm sorry, I am just not sold on the results being an on-target effect.

Reviewer #3 (Remarks to the Author):

Remaining Concerns

The authors need to clearly acknowledge the limitation of the PLN-ASO approach and the required repeated administrations besides the minor point added in the Discussion.

The limitation regarding the use of the PLN-R14del model in the homozygous state is a remaining concern. The authors argue that “heterozygous mice only present the first signs of cardiomyopathy at the age of 18 months: impaired cardiac contractile function, increased myocardial fibrosis and the presence of PLN protein aggregates, thus such a study would span over 2 years”. However, this model resembles the human R14del-PLN patients and has a clear advantage over the homozygous model used here.

In the new data on Ca-cycling in adult cardiomyocytes: Amplitude, upstroke velocity and decay velocity are increased. However, the time to base 50% and the time to base 75% were not significantly altered. Actually, the Time to base 50% appears to be prolonged by PLN-ASO (although not statistically different).

A previous question that is still remaining is: Is SR Ca-load increased by PLN-ASO?

In the R14del-model: there are 4 doses at weeks 3-6 and then assessment is done a week later (at 7 weeks of age). Are there any Ca-defects that PLN-ASO targets and restores in this model?

The authors indicate that “These findings provide further support to the idea that a reduction of toxic PLN protein aggregates by PLN-ASO might contribute to the significant benefits observed after treatment.” However, a question regarding improvement of Ca-defects has not been addressed.

The remaining concerns have been partially addressed in the responses to this reviewer’s comments but they do not appear incorporated in the revised text. Please indicate the corresponding revisions in the revised manuscript as these will provide a better understanding of the current rationale and obtained results. Remaining points include but not limited to: rationale of the study to further decrease the already reduced PLN levels in the R14del-PLN mouse (the authors acknowledge, that this is not caused by transformation of PLN into aggregates); the clarifications regarding the data in Supplemental Fig 5 and Fig 2c (soluble vs insoluble PLN); the concerns regarding Suppl Fig 10 on fibrosis, LV contractility deterioration and ECG as well as the rest of the previously raised concerns.

REVIEWER COMMENTS

Reviewer #1 (Remarks to the Author):

The authors have properly addressed all relevant concerns. They added new data in agreement with the reviewers' suggestions I consider that the revised version of the manuscript is improved.

I have no further concerns.

We thank the reviewer for the kind words.

Reviewer #2 (Remarks to the Author):

I appreciate the authors response.

There have been hundreds of similar papers published claiming ASO-mediated effects in extra-hepatic tissues. The work in the CNS has been convincing, other tissues, not so much. The controls used here remain insufficient to distinguish this work from many other papers. Action in the heart needs to be treated as a remarkable result that needs more than routine support. I cannot support this manuscript but I will understand if editors feel otherwise. It is certainly true that similar papers are routinely published.

There would need to be at least one new experiment using multiple control and multiple target-directed oligonucleotides in vivo to set this submission apart. I'm sorry, I am just not sold on the results being an on-target effect.

We respectfully disagree with the reviewers' comment. Phospholamban is almost exclusively expressed in cardiomyocytes, with minor expression at extra-cardiac sites like skeletal and smooth muscle cells. These data show the PLN-ASO approach is highly specific as multiple ASOs have been evaluated in multiple species (mouse, rat and human) with consistent effects. The beneficial effects observed are as expected for a therapy interfering with the PLN/SERCA interaction in cardiomyocytes. We therefore consider it highly unlikely that the improvement in cardiac structure, function and survival are caused by an off-target effect of any particular ASO sequence or a class effect of the ASOs. This is further strengthened by the finding that these effects are observed when compared to control ASOs (in rat MI and mouse MLP KO models) and that off-target effects of ASOs are generally considered to be detrimental (inflammation, renal dysfunction) instead of beneficial for cardiac function. Also, 2 unique ASO sequences were used in the MLP KO studies with consistent effects observed. Although we feel the utilized control groups are appropriate, an additional control ASO group in the R14del model would have added value and have therefore mentioned this as a limitation in the discussion:

"We do not have data on a scrambled ASO control in the PLN R14del mice study as we prioritized maximizing the group sizes for either vehicle or PLN-ASO treatments. Future studies will need to determine whether any ASO off-target effect can influence R14del aggregate formation or aggregate toxicity."

Reviewer #3 (Remarks to the Author):

Remaining Concerns

The authors need to clearly acknowledge the limitation of the PLN-ASO approach and the required repeated administrations besides the minor point added in the Discussion.

We consider repeated dosing for a PLN targeted therapy an asset of the current ASO approach as repeated dosing will allow careful titration of PLN suppression to safely achieve maximally efficacious improvements in SERCA2a function vs. a non-modifiable dosing approach like gene therapy. This would be especially important considering the potential of the on-target safety concern of the arrhythmogenic potential of increased SR Ca²⁺ leak following excessive SERCA2a activity. For practical reasons, a single dose therapy might be preferred for chronic diseases like heart failure, however, given current ASO treatments in the clinic are administered weekly, monthly or every 4 months, it is likely that a PLN ASO therapeutic might be administered at least on a weekly or monthly basis. We added a statement below to the limitations section to address this point.

"There are several limitations to our approach. As ASOs do not permanently affect gene expression, as might be achieved with gene therapy, repeated subcutaneous administrations will be required for chronic treatment. We feel this is a beneficial attribute, as repeated dosing will allow careful titration of PLN suppression to safely achieve maximally efficacious improvements in SERCA2a function. Like many traditional therapeutics, a therapeutic index can be defined with a margin of safety. This would be especially important considering the potential of the on-target safety concern of the arrhythmogenic potential of increased SR Ca²⁺ leak following excessive SERCA2a activity."

The limitation regarding the use of the PLN-R14del model in the homozygous state is a remaining concern. The authors argue that "heterozygous mice only present the first signs of cardiomyopathy at the age of 18 months: impaired cardiac contractile function, increased myocardial fibrosis and the presence of PLN protein aggregates, thus such a study would span over 2 years". However, this model resembles the human R14del-PLN patients and has a clear advantage over the homozygous model used here.

We agree that ideally, we would have data in heterozygous mice as well. Although it has to be stated that the heterozygous mice do not fully represent the human patients as no DCM phenotype is observed (Eijgenraam et al. Sci Reports, 2020). We now mention this as a limitation in the manuscript:

"The studied PLN R14del mouse model is homozygous, and observed effects might thus not be directly translatable to human heterozygous PLN R14del carriers."

In the new data on Ca-cycling in adult cardiomyocytes: Amplitude, upstroke velocity and decay velocity are increased. However, the time to base 50% and the time to base 75% were not significantly altered. Actually, the Time to base 50% appears to be prolonged by PLN-ASO (although not statistically different).

These data were included in the original submission. The observation is correct. Both the amplitude and speeds are increased in similar magnitude. This results in a similar shaped calcium flux, but with a larger total flux in PLN-ASO treated cells. Therefore, the total calcium flux duration and thus the time to 50% of decay or 75% of decay are unaltered, which can be seen in figure S16A, and below. This is expected as the calcium flux duration is depended on the frequency, which was kept at 1Hz for all conditions. The numerical increase in time to base 50% is explained by the slightly larger increase in amplitude compared to velocities. This is now further clarified in the results section:

"Whole cell calcium flux recordings of adult cardiomyocytes isolated from *Cspr3/Mip*^{-/-} mice after 4 weeks of treatment with PLN-ASO versus control in vivo (Fig.3a), showed significantly enhanced amplitude and upstroke- and decay velocity, without changes in calcium flux durations due to pacing at 1 Hz (Fig. S16)."

A previous question that is still remaining is: Is SR Ca-load increased by PLN-ASO?

This is a good question and related to the previous one. In an elegant in vitro study, Li, Kranias and Bers studied both wild-type and PLN knockout cardiomyocytes (Li, Kranias and Bers, Am J Physiol 1998). In wild-type mouse cardiomyocytes, 90% of cytosolic Ca²⁺ is transported by SERCA to the SR during relaxation. In PLN knockout cardiomyocytes, this increases to 96%. This results in a 37% increase of SR Ca²⁺ content, when the Na/Ca²⁺ exchanger is blocked, and associates with higher calcium decay velocities. We observe a similar Ca²⁺ flux pattern with PLN ASO treatment, indicating

an increased SR Ca²⁺-load with reduced PLN protein levels: an increased Ca²⁺ decay velocity, and an increased Ca²⁺ amplitude; however, we did not quantify the SR Ca²⁺ load. We now mention this as a limitation:

"Calcium fluxes were assessed using relative fluorescence signals, not allowing the absolute quantification of intracellular, or sarcoplasmic reticulum, calcium levels. Future studies will determine the extent of SR Ca²⁺ load following PLN inhibition, and whether such effects occur similarly in rodents large mammals and PLN R14^{D/D}."

In the R14del-model: there are 4 doses at weeks 3-6 and then assessment is done a week later (at 7 weeks of age). Are there any Ca-defects that PLN-ASO targets and restores in this model?

The authors indicate that "These findings provide further support to the idea that a reduction of toxic PLN protein aggregates by PLN-ASO might contribute to the significant benefits observed after treatment." However, a question regarding improvement of ca-defects has not been addressed.

We know that sarcoplasmic reticulum protein folding and processing is highly dependent on local Ca²⁺ levels (Mekahli et al. 2011). This pathway was one of the most differentially expressed pathways between PLN-ASO treated and vehicle treated PLN R14del mice. However, no other Ca²⁺ related pathways were found to be altered. In addition, there is no consensus in the literature on the effects of PLN R14del on Ca²⁺ cycling, with some studies reporting super-inhibition of the mutant PLN on SERCA and others reporting a normal interaction or a non-inhibitory effect. We have no data on cytosolic or sarcoplasmic Ca²⁺ levels in this specific model and have now discussed this in the results and mentioned it in the discussion as a limitation:

"Additionally, we observed a durable downregulation of genes involved in the unfolded protein response in PLN-ASO treated PLN R14^{D/D} mice (Fig. S8f). These findings provide further support to the idea that a reduction of toxic PLN protein aggregates by PLN-ASO might contribute to the significant benefits observed after treatment. Sarcoplasmic reticulum protein processing is dependent on local Ca²⁺ levels²⁸, but no further indication of altered Ca²⁺ related pathways was observed."

"Future studies will need to determine the extent of SR Ca²⁺ load following PLN inhibition, and whether such effects occur similarly in rodents, large mammals and PLN R14^{D/D}."

The remaining concerns have been partially addressed in the responses to this reviewer's comments but they do not appear incorporated in the revised text. Please indicate the corresponding revisions in the revised manuscript as these will provide a better understanding of the current rationale and obtained results. Remaining points include but not limited to: rationale of the study to further decrease the already reduced PLN levels in the R14del-PLN mouse (the authors acknowledge, that this is not caused by transformation of PLN into aggregates); the clarifications regarding the data in Supplemental Fig 5 and Fig 2c (soluble vs insoluble PLN); the concerns regarding Suppl Fig 10 on fibrosis, LV contractility deterioration and ECG as well as the rest of the previously raised concerns.

We now incorporated the clarifications in the manuscript as follows.

Study rationale for further decreasing already reduced PLN levels. (Line 146 – 148)

"A mouse model carrying the PLN R14del pathogenic variant in both alleles (PLN R14^{D/D}) recapitulated all common features of the human phenotype observed in heterozygous carriers, yet in an accelerated fashion, with rapid development of DCM, myocardial fibrosis, and PLN protein aggregates resulting in premature death at the age of 8-9 weeks.²⁷ This phenotype is observed despite endogenous downregulation of PLN RNA and protein expression in untreated PLN R14^{D/D} mice compared to wild-type mice²⁷. We hypothesize this phenotype to be caused by the residual expression of mutant PLN, we aim to further reduce this expression utilizing the PLN-ASO."

Soluble vs. insoluble PLN. (Line 157)

"This resulted in 42% reduction of cardiac Pln mRNA ($\pm 6\%$, P -value=0.0153, Fig. 2b) and 50% reduction of total (both soluble and insoluble/aggregated) PLN protein levels ($\pm 18\%$, P -value=0.0142) at treatment week 4 (T4) and 7 weeks of age (Fig. 2c and S2)."

(Line 162 – 164)

"Immunofluorescence showed a lower abundance of PLN protein aggregates in PLN-ASO treated mice (Fig. 2d). Additionally, RIPA-insoluble protein fractions of PLN R14D/D mice hearts contained more PLN protein as compared to wild-type mice, indicating aggregated PLN protein complexes (Fig. S3-S5)."

Deterioration of cardiac structure and function during follow-up and the interpretation of follow-up PLN protein and mRNA levels. (Line 205 – 210)

"Age-matched vehicle treated control PLN R14^{D/D} mice were not available for these analyses as they all died before the age of 9 weeks, which hampers accurate interpretation of follow-up PLN mRNA and protein analyses in PLN-ASO treated mice. Based on known PLN ASO pharmacokinetics, this dosing regime was chosen not to keep PLN mRNA and protein levels knocked down at steady state, but to allow a slow incline of PLN levels over time and therefore being insufficient to prevent disease progression. PLN-ASO treated PLN R14^{D/D} mice slowly progressed to HF, with first signs of cardiac disease at T19, which corresponds to the age of 22 weeks (Fig. S10)."

Super-rescue of calcium cycling in MLP knockout. (Line 396 – 401)

"Although we did not observe impaired Ca²⁺ fluxes in the Csrp3/Mlp^{-/-} cardiomyocytes, PLN-ASO improved Ca²⁺ handling as hypothesized. Correspondingly, PLN-ASO normalized cardiac function and dimensions in vivo, comparable to earlier results describing the prevention of the Csrp3/Mlp^{-/-} cardiac phenotype when combined with the genetic deletion of Pln or DWORF, a SERCA2a activator, overexpression.^{8,31}"

No rescue of cardiac relaxation in post-MI. (Line 411 – 414)

"Cardiac relaxation was not improved, potentially highlighting the complexity of diastolic relaxation and its dependence on cardiomyocyte morphology and cardiac tissue composition alongside diastolic Ca²⁺ uptake.^{45,46}"

REVIEWERS' COMMENTS

Reviewer #3 (Remarks to the Author):

All remaining comments have been adequately addressed.